# IPB-MSA&SO$_4$: a daily 0.25° resolution dataset of In-situ Produced Biogenic Methanesulfonic Acid and Sulfate over the North Atlantic during 1998–2022 based on machine learning

Karam Mansour[1,2], Stefano Decesari[1], Darius Ceburnis[3], Jurgita Ovadnevaite[3], Lynn M. Russell[4], Marco Paglione[1], Laurent Poulain[5], Shan Huang[5,*], Colin O'Dowd[3] and Matteo Rinaldi[1]

[1] Italian National Research Council, Institute of Atmospheric Sciences and Climate (CNR-ISAC), Bologna 40129, Italy
[2] Oceanography Department, Faculty of Science, Alexandria University, Alexandria 21500, Egypt
[3] School of Natural Sciences, Ryan Institute Centre for Climate and Air Pollution Studies, University of Galway, Galway, Ireland
[4] Scripps Institution of Oceanography, University of California, San Diego, La Jolla, CA, USA
[5] Leibniz Institute for Tropospheric Research, Leipzig, Sachsen, 04318, Germany
[*] Now at Institute for Environmental and Climate Research (ECI), Jinan University, Guangzhou, China

*Correspondence to*: Karam Mansour (k.mansour@isac.cnr.it) & Matteo Rinaldi (m.rinaldi@isac.cnr.it)

**Abstract.**

Accurate long-term marine-derived biogenic sulfur aerosol concentrations at high spatial and temporal resolutions are critical for a wide range of studies including climatology, trend analysis, model evaluation, accurate investigation of their contribution to aerosol burden, or to elucidate their radiative impacts and to provide boundary conditions for regional models. By applying machine learning algorithms, we constructed the first, publicly available, daily gridded dataset of in-situ produced biogenic methanesulfonic acid (MSA) and non-sea-salt sulfate (nss-SO$_4^=$) concentrations covering the North Atlantic Ocean. The dataset is of high spatial resolution of 0.25° × 0.25°, spanning 25 years (1998–2022), far exceeding what observations alone could achieve both space- and time-wise. The machine learning models were generated by combining in-situ observations of sulfur aerosol data at Mace Head research station, west coast of Ireland, and from NAAMES cruises in the NW Atlantic, with the constructed sea-to-air dimethylsulfide flux (F$_{DMS}$) and ECMWF-ERA5 reanalysis datasets. To determine the optimal method for regression, we employed five machine learning model types: support vector machines, decision tree, regression ensemble, Gaussian process, and artificial neural networks. A comparison of the mean absolute error (MAE), root mean square error (RMSE), and coefficient of determination (R$^2$) revealed that the Gaussian process regression (GPR) was the most effective algorithm, outperforming the other models in simulating the biogenic MSA and nss-SO$_4^=$ concentrations. For predicting daily MSA (nss-SO$_4^=$), GPR displayed the highest R$^2$ value of 0.86 (0.72) and the lowest MAE of 0.014 (0.10) µg m$^{-3}$. The GPR partial dependence analysis suggests that the relationships between predictors and MSA and nss-SO$_4^=$ concentrations are complex rather than linear. Using the GPR algorithm, we produced a high-resolution daily dataset of In-situ Produced Biogenic MSA and nss-SO$_4^=$ sea-level concentrations over the

North Atlantic, which we named IPB-MSA&SO$_4$. The obtained IPB-MSA&SO$_4$ data allowed us to analyze the spatiotemporal patterns of MSA, nss-SO$_4^=$, and the ratio between them (MSA:nss-SO$_4^=$). A comparison with the existing CAMS-EAC4 reanalysis suggested that our high-resolution dataset reproduces with high accuracy the spatial and temporal patterns of the biogenic sulfur aerosol concentration and has high consistency with independent measurements in the Atlantic Ocean. The IPB-MSA&SO$_4$ is publicly available at https://doi.org/10.17632/j8bzd5dvpx.1 (Mansour et al., 2023b).

## 1 Introduction

Marine-derived biogenic sulfur aerosol particles exert an important influence on the radiative properties of the atmosphere, both directly by scattering solar radiation and indirectly by modifying cloud properties (Langmann et al., 2008; Charlson et al., 1987). Dimethylsulfide (DMS), a volatile organic compound produced by marine microbes, is the main precursor of biogenic sulfur-containing aerosols in the marine boundary layer (MBL). After being ventilated into the atmosphere, DMS is oxidized to form two of the major secondary marine aerosol species, Methanesulfonic acid (MSA) and non-sea-salt sulfate (nss-SO$_4^=$). Sulfur emitted by marine organisms constitutes 20% (Fiddes et al., 2018) to 40% (Simo, 2001) of the total sulfur burden of the atmosphere. The understanding of the role of MSA and nss-SO$_4^=$ concentrations in Earth's climate is elusive (Mansour et al., 2020a; Hodshire et al., 2019). According to the CLAW hypothesis (Charlson et al., 1987), negative climate feedback is expected to occur if phytoplankton responds to elevated temperature or solar radiation levels by increasing their DMS production, thereby, exerting a cooling effect by increasing the planetary albedo. Indeed, studies confirmed that DMS emissions contribute significantly to stabilizing the Earth's atmosphere (Sanchez et al., 2018; Thomas et al., 2010; Kim et al., 2018; Mahmood et al., 2019; Mansour et al., 2022; Mansour et al., 2020b), while a few others have claimed that the biological control over cloud condensation nuclei (CCN) goes even beyond the CLAW's climatic feedback role of DMS (Quinn and Bates, 2011; Woodhouse et al., 2010; O'Dowd et al., 2004). As a result, biogenic sulfur aerosols play a central role in ocean-atmosphere interactions and regional climate change, and it is critical to parameterize and characterize biogenic MSA and nss-SO$_4^=$ across different sea areas and identify their sources to constrain the past, current and future climate impacts of both species (Hodshire et al., 2019; Gondwe et al., 2003). For instance, MSA observations from Greenlandic ice cores have been used to study the variability of subarctic Atlantic Ocean productivity from decadal to centennial time scales (Osman et al., 2019).

The global aerosol-chemistry-climate general circulation models are used widely to assess the radiative forcing of DMS-derived aerosols. A negative forcing caused by the DMS effect is predicted ranging between −1.7 and −2.3 W m$^{-2}$ (Fiddes et al., 2018; Fung et al., 2022; Thomas et al., 2010; Mahajan et al., 2015). This range is comparable to the positive forcing impact of anthropogenic CO$_2$ emissions (1.83±0.2 W m$^{-2}$) (Etminan et al., 2016). Large uncertainties in DMS forcing estimates (up to ±10 W m$^{-2}$) are partly because models overlook the high-frequency spatial, temporal, and seasonal variability in DMS fluxes (Mansour et al., 2023a; Royer et al., 2015; Mcnabb and Tortell, 2022), and consequent oxidation

products (Riccobono et al., 2014), which are not adequately constrained by the available sparse observations (Bock et al., 2021). This level of uncertainty underlines the need for improved parameterizations of natural sulfur aerosol cycling and fluxes at regional scales (Hulswar et al., 2022; Gali et al., 2018; Mahajan et al., 2015), which is essential for determining their impact on climate. Recently, multilinear regression was utilized to simulate monthly MSA over the eastern China seas at a spatial resolution of $1° \times 1°$ (Zhou et al., 2023), concluding that MSA spatial/seasonal patterns exhibit significant variability, which is primarily governed by surface phytoplankton biomass and the atmospheric boundary layer height.

Focusing on the North Atlantic (NA) Ocean, sulfur-containing aerosols, MSA and nss-$SO_4^=$, have been measured at Mace Head sampling station, a coastal area in the eastern NA Ocean, to quantify the contribution of phytoplankton emissions to aerosol mass concentrations in MBL (Rinaldi et al., 2010; Rinaldi et al., 2009; O'Dowd et al., 2004), to assess the long-term seasonal patterns in the chemical composition of submicron aerosol in the different origin of marine air masses (Ovadnevaite et al., 2014), and to identify the oceanic regions acting as the main source of biogenic aerosols (Mansour et al., 2020b). During NAAMES field campaigns, research cruises aimed at comprehending the relationships between ecosystems, aerosols, and clouds (Behrenfeld et al., 2019), Saliba et al. (2020) evaluated the origins and contributions of submicron organic and sulfate components to CCN concentrations in the MBL. They concluded that the DMS-derived secondary nss-$SO_4^=$ enhanced hygroscopicity, particle size, and CCN concentrations by 5–66%, especially in the spring, highlighting the importance of phytoplankton produced DMS emissions for the CCN budget in the NA (Mansour et al., 2022; Mansour et al., 2020b; Sanchez et al., 2018). However, it is currently challenging to effectively investigate climatology, long-term trends and climate forcing of biogenic sulfur compounds, as well as validate inherent model outputs, since there is a lack of high-time resolution data on these compounds.

In this study, we present the first high-resolution and long-term daily gridded time series of freshly formed In-situ Produced Biogenic Methanesulfonic Acid and nss-Sulfate (IPB-MSA&SO4) concentrations over the NA ocean at $0.25° \times 0.25°$ spatial resolution. The data covers 25 years from 1998 to 2022 with the possibility of future updating year by year. The dataset is a unique and novel product that in fact extends the space and time representativeness of atmospheric in-situ observations of marine aerosol chemical properties over the NA Ocean, by exploiting the potential of machine learning. The dataset represents the sea-level concentrations of MSA and nss-$SO_4^=$, in each grid point of the domain, resulting from the interplay between precursor emissions and local atmospheric conditions. We created the IPB-MSA&SO4 dataset using in-situ MSA and nss-$SO_4^=$ data measured at Mace Head (MHD) site and from NAAMES cruises, the gridded dataset from the ECMWF-ERA5 together with the reconstructed $F_{DMS}$ (Mansour et al., 2023a) as input data. To achieve this aim, we employed machine learning (ML) approaches: support vector machines (SVM), decision tree (DT), regression ensemble (RE), Gaussian process regression (GPR), and artificial neural networks (ANN). ML has been applied in a variety of scientific areas for model approximation, experiment design, and multivariate regression of oceanic and atmospheric complex systems, however, no prior applications to MSA and nss-$SO_4^=$ prediction have been published, to our knowledge. During model training, we

evaluated the various possible kernel functions and hyperparameters in each ML type (details in Table 1), employing the 5-fold cross-validation strategy to select the best-performing (optimal) function capable of properly predicting MSA and nss-$SO_4^=$. The partial dependence analysis is also used to assess the effect of different predictors on the modeled MSA and nss-$SO_4^=$. Furthermore, we investigate the annual and monthly spatial distributions of MSA, nss-$SO_4^=$ and the ratio between them (MSA:nss-$SO_4^=$) to examine the evolution of MSA and nss-$SO_4^=$ in the different regions of the NA domain from 1998 to 2022. The output data (IPB-MSA&SO4) from this study should be useful for filling the data gap, particularly for the NA, and be applicable to a variety of investigations, such as climatology, trend analysis, model evaluation, radiative impacts, and providing boundary conditions for regional models.

## 2 Study domain and data sources

### 2.1 Study area and measuring sites

The study area extends from 20° to 66° N and from 72° W to the prime meridian (Fig. 1) covering the NA Ocean. The key climate-relevant features in the study domain are the Gulf Strem, its northern extension towards Europe known as the North Atlantic Current (NAC), and the cyclonic subpolar gyre (SPG) (Rhein et al., 2011). The Gulf Stream is a warm Atlantic Ocean flow that begins in the Gulf of Mexico and moves through the Straits of Florida before continuing up the eastern coast of the United States (Buckley and Marshall, 2016). These warm northward-flowing waters meet the cold southward-flowing waters of the Labrador Current and the western boundary current of the cyclonic subpolar gyre, ultimately turning east and heading toward Northwest Europe as the NAC. The NAC then splits into multiple branches that enter the subpolar gyre, one of which passes via the Iceland Basin and the other through the Rockall trough (Fratantoni, 2001). The NA SPG extends from 45° N to around 65° N and comprises the sills between Greenland, Iceland, the Faroe Islands, and Scotland. Such circulation phenomena are crucial for the modulation of the temperate climate of north-western Europe (Marzocchi et al., 2015), and the dynamics of SPG determine the rate of deep and intermediate water formation (sinking dense and cold surface waters through air-sea heat exchanges in the wintertime) particularly in the Labrador Sea (Katsman et al., 2004). Accordingly, they contribute to the regional changes of primary production and the subsequent biogenic emissions in the study domain.

The MHD global atmospheric watch (GAW) research station (53.33° N, 09.90° W) is located on Ireland's west coast (Fig. 1), at about 80 meters from the coastline and 21 m above mean sea level. MHD is the only GAW station in the eastern Atlantic region and is the globally acknowledged clean background western European station, providing key baseline input for intercomparing with levels elsewhere in Europe (Grigas et al., 2017; O'Dowd et al., 2014).

Four shipboard field campaigns were carried out as part of the NAAMES research project (Behrenfeld et al., 2019). The tracks of cruises representing marine conditions during aerosol sampling (Saliba et al., 2020) are shown in Fig. 1. The

measurements cover the periods of November 2015, May–June 2016, September 2017, and March 2018. Behrenfeld et al. (2019) provide a thorough explanation of the NAAMES project's goals, objectives, and atmospheric and oceanic conditions.

## 2.2 Observational data

The long-term submicron sulfur aerosol species atmospheric concentrations (Methanesulfonic acid [MSA] and Sulfate [$SO_4^=$]) from January 2009 to June 2018 measured at MHD were used. The measurements were performed by using the Aerodyne High Resolution- Time of Flight- Aerosol Mass Spectrometer (HR-ToF-AMS). The HR-ToF-AMS (Decarlo et al., 2006) output has a time resolution of ~5-10 minutes and it was operated according to the recommendations by Jimenez et al. (2003), Allan et al. (2003) and Canagaratna et al. (2007). The MSA was derived from the concentration of mass fragment

$CH_3SO_2^+$ (Ovadnevaite et al., 2014). Further information on the MSA measurement can be found in Mansour et al. (2020a). The black carbon (BC) concentrations were measured in-situ at MHD by a multi-angle absorption photometer (O'Dowd et al., 2014) to identify the anthropogenically impacted air masses, as detailed in Section 3.1.1.

High-resolution in-situ shipborne measurements of non-refractory submicron $SO_4^=$ concentrations were measured every 5 min using HR-ToF-AMS during four open-ocean research cruises (NAAMES) in the NW Atlantic [4 campaigns represent

winter (November 2015), late spring (May–June 2016), autumn (September 2017), and early spring (March 2018)] (Saliba et al., 2020). We employ the $SO_4^=$ concentrations, whereas there are no high-resolution MSA datasets available from NAAMES campaigns, during periods that were largely marine aerosol sources which were defined as periods when particle number concentrations <1500 cm$^{-3}$, BC <50 ng m$^{-3}$, 2-days back trajectories originated from the North or tropical Atlantic, and radon concentrations <500 mBq m$^{-3}$ according to Saliba et al. (2020). The measured $SO_4^=$ from AMS excludes refractory

particles that likely contain the majority of sea-salt sulfate which is therefore approximately equivalent to nss-$SO_4^=$ (Frossard et al., 2014).

## 2.3 Air mass back-trajectories

The Air Resources Laboratory (ARL) of the National Oceanic and Atmospheric Administration (NOAA) developed the Hybrid Single-Particle Lagrangian Integrated Trajectory (HYSPLIT4) model (Rolph et al., 2017; Stein et al., 2015), which is

used to calculate the air mass back-trajectories (BTs). The archived Global Data Assimilation System (GDAS1) (1° × 1°) of the National Centers for Environmental Prediction (NCEP) was used as a driver of the trajectory calculation (ftp://arlftp.arlhq.noaa.gov/pub/archives/gdas1). We run the model at the MHD sampling station as a fixed source location and throughout the NAAMES cruises as a moving source location. The starting height is set to be 100 m above ground level and the backward time is 3 days with an interval of 1 h along each entire trajectory track. The schematic diagram of BTs

calculation is shown in Fig. S1. The arrival frequency of BTs at MHD is 3h (eight tracks a day) covering the period from 01-Jan-2009 to 30-Jun-2018 and of NAAMES is hourly (twenty-four tracks a day) covering the time of the four campaigns identified as marine periods (Saliba et al., 2020).

### 2.4 Dimethylsulfide flux data

The seawater DMS is the primary contributor to biogenic sulfur aerosol in the atmosphere. For this reason, we use the sea-to-air DMS flux ($F_{DMS}$) as a predictor of MSA and nss-$SO_4^=$ concentrations. Mansour et al. (2023a) used an ML predictive algorithm based on Gaussian process regression (GPR) to simulate the distribution of daily seawater DMS concentrations and related $F_{DMS}$ in the NA areas from 35° to 66° N and from 0° to 55° W at 0.25° × 0.25° spatial resolution. We extended the GPR model within the NA to encompass the NAAMES measurements, which are essential because they cover the western most section of the study area. Fig. S2 displays the main differences between the two domains. Simply, the GPR was trained once more, utilizing the same approach of Mansour et al. (2023a), with a higher number of data points and yielded an enhanced $R^2$ value up to 0.77 on the independent test dataset. The daily sea-to-air $F_{DMS}$ was calculated using the gas transfer velocity (Goddijn-Murphy et al., 2012) and the DMS derived from GPR predictions. For more details about the data product, we refer the reader to Mansour et al. (2023a).

### 2.5 Meteorological data

The ECMWF-ERA5 reanalysis data (Hersbach et al., 2020) were downloaded to extract the meteorological parameters used as predictors of MSA and nss-$SO_4^=$ in the ML models. ERA5 provides estimates for the hourly state of the atmosphere, worldwide, with spatial resolution 0.25° × 0.25° at the surface and different pressure levels. From the global domain, we extracted multiple atmospheric components including air temperature at 2m above sea level (AT) and surface net short-wave radiation flux (SRF) as representative of thermal heating, and the relative humidity (RH) as representative of water vapor abundance in the atmosphere. To represent the dispersion of aerosol particles in the troposphere and the wet removal through the below-cloud scavenging process, the boundary layer height (BLH) and the precipitation rate (PR) were utilized, respectively.

## 3 Methods

### 3.1 Data preparation

In this Section, we describe the preparation of predictors and responses that were used to train, cross-validate, and generate the ML models.

#### 3.1.1 Air mass selection

In previous studies (Mansour et al., 2020b; O'Dowd et al., 2015; Ovadnevaite et al., 2014), BC concentration was often considered as a useful tool to select clean marine air masses excluding inputs from continental emissions or ship trails. In this study, we still relied on BC measurements as a precious tool to identify and exclude anthropogenically impacted air

masses, but we also developed a more complete approach aimed at identifying air masses characterized by a high degree of contact with the ocean surface. This was necessary in order to select, from the in-situ observations, data points representing almost entirely oceanic sources to provide the best dataset for training the ML models.

The retention ratio of the air mass over the ocean ($R_O$) was calculated to determine whether an air mass (identified by BT track) arriving at the MHD sampling station or at the ship location, in the case of shipborne measurements, was primarily from the NA region or not. We used 3-day BTs arriving 100 m above the MHD sampling station and NAAMES tracks. The BTs tracks at the MHD arrival point were calculated 8 times per day, whereas it was 24 times per day at NAAMES measuring points, considering only the measurements classified as marine periods (Saliba et al., 2020). The $R_O$ has been calculated for each track as:

$$R_O = \frac{\sum_{i=1}^{N_{Ocean}} e^{\frac{-t_i}{72}}}{\sum_{i=1}^{N_{Total}} e^{\frac{-t_i}{72}}} \tag{1}$$

where $N_{Total}$ is the total number of trajectory endpoints which is equal to 73 (arrival point + 72 backward hours). $N_{Ocean}$ is the total number of trajectory endpoints passing over the ocean, while $t_i$ is the backward tracking time with the unit of an hour spanning the values from 0 to 72. Because air mass diffusion and particles deposition potentially occur during the air mass transport, a weighting factor $e^{-t_i/72}$ related to tracking time has been introduced. The weighting factor takes the values from 1 (at the arrival point) up to 0.37 (farthest point), hence, the oceanic areas far from the arrival point, corresponding to longer backward tracking time, have a weaker influence than areas closer to the sampling point. As a result, a higher $R_O$ value implies that oceanic emissions have a greater influence on the air mass and that the source region is more likely to be the ocean. Other studies have used similar methods to characterize air mass source regions. For example, Zhou et al. (2021) studied the contribution of non-marine MSA sources in the coastal East China Sea and the Gulf of Aqaba by characterizing the land air masses. Rinaldi et al. (2021) used a combination of low-travelling air mass BTs and satellite ground-type maps to investigate the effect of ground conditions (sea ice, snow, seawater, and land) on air samples at Ny-Ålesund station in the Arctic Ocean.

Because oceanic air masses crossing the NA can pass above the BLH, its connection to local sea surface processes such as marine biogenic emission and subsequent atmospheric reactions may be significantly weaker. To address this issue, Eq. 2 was used to calculate the retention ratio of an ocean air mass within the marine boundary layer ($R_B$).

$$R_B = \frac{\sum_{i=1}^{N_{Below}} e^{\frac{-t_i}{72}}}{\sum_{i=1}^{N_{Ocean}} e^{\frac{-t_i}{72}}} \tag{2}$$

where $N_{Ocean}$ is the total number of trajectory endpoints located over the ocean (i.e., marine endpoints) and $N_{Below}$ is the number of marine endpoints which have an altitude below BLH. The higher the $R_B$ value, the more airflow over the ocean is confined to the MBL. The BLH datasets at each endpoint were extracted from the hourly ERA5 dataset.

The total number of BTs tracks arriving at MHD during the period from Jan-2009 to Jun-2018 is 27,744 (3468 days × 8 tracks per day). We counted the number of endpoints of all BTs in each 1° × 1° grid cell and normalized them to the maximum value to find the percentage of endpoints for all grid cells (Fig. S3). The larger density of BTs endpoints is concentrated over the NA oceanic region, indicating that the main source regions for air masses transported to MHD sampling stations are most likely oceanic. At MHD, we investigated how MSA (a marine biogenic tracer) responds to

change in BC (a tracer of anthropogenic input) as seen in Fig. S4, by considering hourly data simultaneous to the arrival time of BTs (*i.e.*, 8 times a day). We found that MSA tends to fluctuate minimally when BC is less than 15 ng m$^{-3}$ (slope = 0.05), whereas MSA tends to rise slightly when BC exceeds 15 ng m$^{-3}$ (slope = 0.28). Such cases with hourly BC concentrations <15 ng m$^{-3}$ were classified as representative of marine conditions, that are likely not influenced by anthropogenic sources. To constrain the impact of marine biogenic emissions and meteorological parameters on MSA and nss-SO$_4^=$, air masses were

included in this analysis only if they were characterized by $R_O + R_B \geq 1.75$, meaning that the air mass had a high degree of contact with the ocean surface within the last 3 days (Fig. S4). Indeed, considering the above condition, an air mass must have at least $R_O$ equal to 0.75 and in such case the track must be traveling 100% of the time below the BLH. By introducing the criterion of $R_O + R_B \geq 1.75$, approximately 72% of the BTs tracks were considered. This reflects the significance of the MHD research station for studying NA biogenic emissions, and the frequency with which it is impacted

by MBL air masses (Grigas et al., 2017; O'Dowd et al., 2014). After considering the BC threshold (<15 ng m$^{-3}$) and conservatively removing all the observations done when the BC data were unavailable (instrument downtime), 9211 (33% of the total) tracks were classified as representative of marine conditions (selected marine BTs frequency is presented in Fig. S5).

Regarding the NAAMES measurements, the total number of calculated BTs tracks was 832 (Fig. S6) during background

marine conditions, identified by Saliba et al. (2020). In this study, we kept 660 tracks (Fig. S7) of the above 832 as representative samples of marine conditions during NAAMES cruises by limiting the analysis to hourly samples with $R_O + R_B \geq 1.75$.

### 3.1.2 Predictors extraction along back trajectories

In order to train the ML models, it was necessary to associate each observed MSA and nss-SO$_4^=$ data point with the

240 corresponding potential predictors. The potential predictors (F$_{DMS}$, AT, SRF, RH, BLH and PR) were extracted at each endpoint of the BTs associated with each of the selected clean marine observational data points (see Section 3.1.1), inside the oceanic region within 20−66 °N and 0−72 °W (Fig. S1). The extracted predictor values were then averaged along each

marine BT track, providing the most representative picture of the conditions (air mass history) that led to the formation of the observed sulfur aerosol concentrations. The few endpoints over land or crossing above the BLH were eliminated.

The Pearson's correlation coefficients between the potential predictors and observational MSA and nss-SO$_4^=$ data were compared, considering different BT lengths of 1, 2 and 3 days, to assess which BT length was more representative of the time scale of sulfur aerosol formation processes. As seen from Table 2, both MSA and nss-SO$_4^=$ correlate better with F$_{DMS}$ considering a 3-day BT length. Similarly, the majority of the other predictors, except for AT, tended to maximize their correlations considering 2 or 3 days of BT length. Ultimately, we considered for each predictor the BT length that

maximized the correlation coefficient for the analyses in the present study.

### 3.1.3 Responses at measuring sites

Hourly nss-SO$_4^=$ at MHD and from NAAMES campaigns as well as MSA at MHD, measured concurrently with the selected marine BTs (Section 3.3.1), were used to build ML models. A total of 6162 (6920) data points for MSA (nss-SO$_4^=$) were obtained. Further, we also applied 0.1 and 99.9 percentiles lower and upper thresholds filter to remove the extremely low and

high values that could bias the ML models training and cross/validation. This helped to identify and remove outliers in each dataset, thereby reducing the number of data points to 6150 (6905) for MSA (nss-SO$_4^=$) (~0.2 % of data points were rejected). Details of the MSA and nss-SO$_4^=$ percentile thresholds, along with the amount of data before and after applying the filters are given in Table 3. The hourly data after cleanup is used for training/ cross-validation and testing of ML models.

### 3.2 Machine learning models

The methodological flowchart of the present study is shown in Fig. 2. The core of the framework is using the supervised ML regression techniques to build predictive models for estimating the atmospheric concentrations of biogenic MSA and nss-SO$_4^=$ (responses) from independent variables (predictors). Predictors include the sea-to-air F$_{DMS}$ and meteorological parameters that control the aerosol concentration in the MBL. We used multilinear regression to assess the contribution of each predictor to MSA and nss-SO$_4^=$ variations. Initially, we ran the multilinear regression model using the total of the

potential six predictors: F$_{DMS}$, AT, SRF, RH, BLH and PR. Secondly, we applied the multilinear regression models by eliminating one predictor each time. Each independent variable's contribution to R$^2$ is the reduction in total R$^2$ when that variable is eliminated. The results (Table 4) showed that the six predictors used can explain up to 74% (53%) of MSA (nss-SO$_4^=$) variance. Such predictors tend to contribute differently to MSA and nss-SO$_4^=$. SRF, F$_{DMS}$ and BLH are the most effective parameters for MSA (explaining up to 64 % of the variability), while SRF, AT and F$_{DMS}$ are the most influential on

nss-SO$_4^=$ (explaining up to 44 % of the variability). RH has a minor contribution to the MSA and nss-SO$_4^=$ variance. To know if a predictor contributes significantly to the explained variance, we performed the analysis of variance (ANOVA) on the implemented multilinear regression model. The ANOVA revealed that all the tested predictors have statistically significant

($p < 0.05$) contributions to MSA and nss-$SO_4^=$. For these reasons, we applied the ML models using all of the potential six predictors.

The datasets, containing the corresponding predictors and each one of the responses (MSA and nss-$SO_4^=$) separately, were split randomly into two subsets, defined as the training/cross-validation set and the test/evaluation set for each response. The training/cross-validation sets include 80% of the total points ($n$ = 4920 for MSA and $n$ = 5524 for nss-$SO_4^=$), while the test/evaluation sets comprise the remaining 20% ($n$ = 1230 for MSA and $n$ = 1381 for nss-$SO_4^=$). To improve ML algorithms' accuracy and protect against overfitting, a k-fold cross-validation strategy, with $k = 5$ was used, as this has been shown to

provide maximal model prediction robustness and minimal bias (Rodriguez et al., 2010; Fushiki, 2011). The k-fold cross-validation is a procedure used to estimate the skill of the model on new data and generally results in a less biased estimate of the model skill. The number k-fold refers to how many groups a given data sample is to be split into. In this study where $k = 5$, the training/cross-validation dataset randomly was further divided into 5 folds of roughly equal size. At each trial, one group is designated as a holdout or validation dataset, while the remaining four groups are designated as training data (Fig.

2). The model is then fit on the training set (4 folds) and evaluated on the validation set (last fold), and the average evaluation measures (accuracy) on the validation subsets of the five iterations are reported. To better examine the model's repeatability on a new independent dataset, the generated models were evaluated on the test data that was not included in the model construction.

    Five types of ML models were trained/cross-validated and evaluated to identify the best-performing model in estimating

sulfur aerosol concentrations (MSA and nss-$SO_4^=$). The ML algorithms are SVM, DT, RE, GPR, and ANN. These are the most common types of algorithms, but still, there are subtypes where advanced options and optimizations in the model can increase the performance and resilience of the algorithms. In general, each supervised ML model performs differently and has various strengths and shortcomings. Finding the proper ML algorithm is largely based on trial and error; even experienced data scientists cannot anticipate if an algorithm will work without testing it. Thus, understanding the

fundamentals of various ML algorithms and their applicability in diverse applications is critical (Sarker et al., 2019). As a result, initially, we assessed 20 algorithms belonging to the aforementioned five types and chose the most fitted from each type (Table 1), as detailed in the following Sections.

### 3.2.1 Support vector machines (SVM)

    SVM is a powerful mathematical model based on the statistical learning theory (Vapnik, 2013) that can be used either for

classification or regression analysis. In recent decades, SVM demonstrated high prediction accuracy in a wide range of regression problems in fields such as oceanography, meteorology, and atmospheric sciences (Lins et al., 2013; Sachindra et al., 2018; Shabani et al., 2020; Shrestha and Shukla, 2015; Fan et al., 2018). The SVM model estimates the regression using a series of kernel functions that are capable of implicitly converting the original, lower-dimensional input data to a higher-

dimensional feature space. To achieve the best prediction accuracy for MSA and nss-$SO_4^=$, we assessed the SVM different kernel functions such as linear, polynomial (quadratic and cubic) and Gaussian (Table 1). The Gaussian kernel was adopted by trying various kernel scales, setting them to 0.61 (fine), 2.4 (medium), and 9.8 (coarse). For more information on SVM, the reader is referred to https://www.mathworks.com/help/stats/fitrsvm.html.

### 3.2.2 Decision tree (DT)

The DT model is a non-parametric, non-linear model that generates a structure resembling a tree for classification and regression (Kotsiantis, 2013; Quinlan, 1986). It repeatedly divides the dataset into smaller subsets based on independent features from the input dataset. The split seeks to reduce variability within each group while increasing the variance between subsets. The final tree is made up of decision and leaf nodes. The decision node represents a condition on an attribute, and its branches indicate the conditions' outcomes. For additional information on DT, the reader is directed to https://www.mathworks.com/help/stats/fitrtree.html. The critical parameter in this technique is determining when to terminate the dividing process. In this study, we set up three different minimum leaf sizes (minimum samples to split) to control the number of data that should be in the sub-branch to continue the splitting process, namely 4 (fine tree), 12 (medium tree), and 36 (coarse tree) as seen in Table 1.

### 3.2.3 Regression ensemble (RE)

The ensemble is a technique that employs a collection of DT models (referred to as weak learners or base models), each of which is produced by applying a learning process to a specific problem and then combining them to provide the final prediction (Mendes-Moreira et al., 2012). The performance and accuracy of ensembles are determined by the aggregation of weak learners (Hengl et al., 2018). The well-known types of aggregation are the bagging and boosting methods (Breiman, 2001). In the bagging method (also known as bootstrap aggregating), the base models are generated using random sub-samples drawn from the original dataset with the bootstrap sampling method, where some original examples appear several times while others do not appear at all. On the other hand, the main idea of the boosting method is that it is possible to convert a base model that performs slightly better into one that arbitrarily achieves high accuracy. This conversion is performed by combining the estimations of several predictors. For more information on RE, the reader is referred to https://www.mathworks.com/help/stats/fitrensemble.html.

### 3.2.4 Gaussian process regression (GPR)

GPR is a non-parametric technique for solving nonlinear regression problems (Williams and Rasmussen, 1996) which is based on Bayesian theory and statistical learning theory. The accuracy of GPR is dependent on the adopted kernel (covariance) functions (Verrelst et al., 2016). We assessed the different base kernel functions, namely exponential, Matern 5/2, squared exponential, and rational quadratic (Asante-Okyere et al., 2018; Mansour et al., 2023a) to determine the optimal

covariance function that could produce reliable predictions of MSA and nss-$SO_4^-$. For more information on GPR, the reader is referred to Mansour et al. (2023a) and https://www.mathworks.com/help/stats/fitrgp.html.

### 3.2.5 Artificial neural networks (ANN)

ANN is an information processing system, which can be used to understand the complex nonlinear relationship between the response and predictors (Kalogirou, 2001). It consists of interconnected groups of artificial neurons that work in the same way as biological neurons. The ANN structure comprises three distinctive groups called input (corresponds to the predictors), several hidden layers (fully connected), and output (corresponds to the predicted response values). The input introduces data to the ANN model, the hidden layer processes the data, and the results are produced in the output. Further details on ANN can be found at https://www.mathworks.com/help/stats/fitrnet.html. We trained various types of ANN as single-layer (number of fully connected layers = 1), bi-layered (number of fully connected layers = 2), and tri-layered (number of fully connected layers = 3) neural networks as detailed in Table 1.

### 3.3 Evaluation measures

In this study, we use different validation metrics to evaluate the ML models' performance. Each of the metrics is calculated using "residuals". Residuals are the differences between the observed data points $O_i$ and the predicted values $P_i$, where $i = 1, 2, \dots n$. $n$ refers to the number of observations. Better models in predicting the response have residuals close to zero. The average magnitude of the residuals is called mean absolute error (MAE).

$$MAE = \frac{1}{n}\sum_{i=1}^{n}|O_i - P_i| \tag{3}$$

Regression models tend to use the square of the residuals instead of the absolute. The square root of the average of the squared residuals is called root mean square errors (RMSE). A low RMSE is a confidence that your model has relatively few large errors.

$$RMSE = \sqrt{\frac{1}{n}\sum_{i}^{n}(P_i - O_i)^2} \tag{4}$$

The metrics listed in Eqn. 3 and Eqn. 4 can only tell you how a model compares to observations and/or other models. Neither can say whether a model is a good fit for the data objectively. Comparing a model to a simple baseline model is a different approach. This is the motivation behind the use of the coefficient of determination ($R^2$) metric (Eqn. 5). $R^2$ is the relative difference in the total error obtained by fitting a model, so a value between 0 and 1. If a model fits the data well, the model error is small and $R^2$ will be close to 1 and vice versa.

$$R^2 = 1 - \frac{\sum_{i}^{n}(O_i - P_i)^2}{\sum_{i}^{n}(O_i - \overline{O_i})^2} \tag{5}$$

Where $\overline{O_t}$ is the average of observations. The predicted-observed linear slope is the last metric used to evaluate the performance of ML models. It determines the rate of change of the predicted variable concerning the observed variable and should be close to unity for skilled model predictions.

## 4. Results and Discussion

**4.1 Evaluation of ML model performance**

As a first step, we assessed different possible hyperparameters optimization in each type of the five used ML models (SVM, DT, RE, GPR, and ANN) to determine which one has the best fit and lesser errors in sulfur aerosol (MSA and nss-$SO_4^=$) predictability. We chose the best model with the least errors in each type for further evaluation and analysis based on the evaluation measures (RMSE, MAE, and $R^2$). The evaluation measures are summarized in Table S1. The medium Gaussian

SVM which utilizes a Gaussian kernel scale equal to the square root of the number of predictors (= 2.4), displayed better performance. The coarse DT, which sets the minimum sample size to split equal to 36, the ensemble bagged trees (EBT) of a bootstrap aggregated ensemble and the GPR, which employs the rational quadratic kernel, represent the minimum errors. Finally, a medium ANN of layer size 25 with one fully connected layer is selected. The five best-performing (optimal) models have been exported and saved so that they can be used to make new predictions on a new dataset.

Fig. 3a-e and Fig.4a-e present the detailed comparison between observed and predicted MSA and nss-$SO_4^=$, respectively, of the five developed ML optimal models. When compared to the multilinear regression (Table 4), it is clear that ML models, in general, can reconstruct the observations with a markedly higher $R^2$ value, which means that the selected ML approaches capture much more of the observed MSA and nss-$SO_4^=$ variability. While the five applied optimal algorithms have quasi-similar measures, the best model is GPR for predicting MSA and nss-$SO_4^=$. For hourly MSA (nss-$SO_4^=$), the GPR achieves the

highest $R^2$ value of 0.79 (0.64) and the least RMSE of 0.362 (0.282) for the cross-validated data (average measures of each validation fold). When extending to the test data, $R^2$ and RMSE reach 0.81 (0.67) and 0.347 (0.272), respectively. The EBT comes second in terms of performance in predicting MSA (nss-$SO_4^=$) with $R^2$ = 0.80 (0.64) of the independent test data. The SVM and ANN achieve a reasonable accuracy with $R^2$ = 0.79 (0.61) and 0.78 (0.60), respectively for MSA (nss-$SO_4^=$) based on the test data. Lastly, based on the hourly test data, the DT shows the lowest, but still respectable, accuracy with $R^2$ = 0.76

for MSA and = 0.57 for nss-$SO_4^=$.

Importantly, the implemented ML models can reconstruct MSA and nss-$SO_4^=$ daily time series characteristics with remarkable consistency between observed and predicted data, except for extremely high and low concentrations. This is mostly due to the low probability of such concentrations in the observed dataset, which inhibits ML models from reconstructing them. The quantitative comprehension of exceptional emission extremes is not addressed in this study;

nonetheless, their occurrence and possible implications deserve to be investigated in future studies. It is worth noting that the

daily averages of MSA and nss-$SO_4^=$ have been calculated from the validation folds and the test set. The MAE of GPR is close to 0.014 (0.100) µg m$^{-3}$ for MSA (nss-$SO_4^=$). The MAE of EBT, SVM, ANN and DT are higher than those of both GPR. According to the $R^2$, the ranking order is the same as for MAE, i.e., GPR outperforms EBT, SVM, ANN and DT in both MSA and nss-$SO_4^=$, notwithstanding the differences in the $R^2$ of the five models are small. An in-depth look at the MAE

and $R^2$ from MHD and NAAMES (Fig. 4; right panels) demonstrates that the ML models perform well in predicting nss-$SO_4^=$ across different datasets. All five models show relatively high values of $R^2$ on the NAAMES dataset. EBT, SVM and ANN have $R^2$ values that are similar and equal to 0.81, whilst GPR has the highest value of $R^2$ reaching 0.87 and DT has the smallest at 0.72. In essence, the performance metrics indicate that GPR always has the highest accuracy and lowest errors, reflecting the robustness of GPR. Therefore, GPR was selected as the optimal regressor for further analysis throughout this

study.

Knowing that the GPR model could be biased due to the inhomogeneous distribution of in situ observations, we assessed the applicability of the GPR model in regions poorly covered by atmospheric observational data (as the central part of the domain) by running the model in a worst-case scenario deployment. In this exercise, we predicted the daily variations of nss-$SO_4^=$ measurements in the westernmost portion of the study area by training the model only with observations from the

405 eastern part of the domain (*i.e.*, data collected at MHD). In this case, MHD data were used for training/cross-validation, while the four NAAMES campaigns were employed as independent test data. The evaluation on the test data (Fig. S8) reveals that GPR can explain 55% of the daily observed nss-$SO_4^=$ variance (MAE = 0.129 µg m$^{-3}$), even in this worst-case scenario and on a limited test dataset ($n$ = 57). This more than acceptable performance of the model supports the reliability of the IPB-MSA&SO4 dataset also in the central part of the NA, where measurements of MSA and nss-$SO_4^=$ are missing. In

addition, Section 4.5 describes the validation of the GPR model for predicting observed MSA concentrations during the Polarstern campaigns, which were not included in either the model training/cross-validation or in the model test.

### 4.2 Partial dependence analysis

The bulk of ML models is called a "black box" since the internal computations inside multiple operational layers in a model are concealed and most systems have only observable inputs and outputs out of the box. The partial dependence analysis

(Friedman, 2001) is used to assess how predictors influence an output by ML model and show whether the relationship between the response and any of the features is linear, monotonic or more complex. The method entails altering one feature and constraining the remaining features to unaltered average values to illustrate the marginal effect of the changed feature on the expected outcome. The partial dependence plots of MSA and nss-$SO_4^=$ as a function of the predictors in the highest-performing GPR model are shown in Fig. 5, indicating that the interactions between predictors and response are complex in

general. MSA and nss-$SO_4^=$ levels tend to rise as $F_{DMS}$ levels rise from 3 to 10 µmol m$^{-2}$ d$^{-1}$. MSA continues to rise with stronger $F_{DMS}$ emission rates (>10 µmol m$^{-2}$ d$^{-1}$), nevertheless, nss-$SO_4^=$ concentration appears independent of $F_{DMS}$ after this threshold. AT exhibits a positive relationship with MSA and nss-$SO_4^=$ concentration in the range of (5–10 °C) and above a

downward trend. RH, which has the least impact on MSA and nss-$SO_4^=$ (Table 4), has an unclear pattern on the MSA and nss-$SO_4^=$ marginal changes. MSA and nss-$SO_4^=$ present a negative dependence on PR as rain is expected to scavenge aerosol particles; nevertheless, at higher levels of PR, nss-$SO_4^=$ concentrations tend to increase. This may be partly linked to enhanced cloudiness, associated to high PR, where the aqueous phase formation of nss-$SO_4^=$ in the MBL may be favored (Zhu et al., 2006; Von Glasow and Crutzen, 2004). This is also in agreement with the enhancement of nss-$SO_4^=$ concentration at high RH. Finally, BLH and SRF are the most straightforward influencing parameters on MSA and nss-$SO_4^=$ levels, with deep BLH resulting in a dilution of their concentrations and high SRF leading to high MSA and nss-$SO_4^=$ levels, as expected for DMS photo-oxidation products.

### 4.3 The IPB-MSA&SO₄ dataset

The GPR model was used to generate the long-term gridded fields of high-resolution (0.25° × 0.25°) MSA and nss-$SO_4^=$ concentrations. At each pixel, a daily time series of MSA and nss-$SO_4^=$ have been generated spanning from 1998 to 2022 (9131 days). The total number of pixels in the entire NA domain is 43840, for a total of 400'303'040 data points. The daily time series of MSA and nss-$SO_4^=$ averaged over the entire NA domain are presented in Fig. S9. The dataset represents the sea-level concentrations of MSA and nss-$SO_4^=$ associated with in-situ production in the MBL derived based on the six selected predictors, which in turn represent the sea-to-air flux of DMS (the precursor) and the meteorological conditions that can mostly affect, in one direction or in the other, the formation of the two products. For this reason, we consider the data to be representative of the concentration of sulfur aerosol species resulting, in each pixel, from the local biogenic emissions in combination with local atmospheric conditions. As such, we called the achieved data product the In-situ Produced Biogenic MSA and nss-$SO_4^=$ (IPB-MSA&SO₄) dataset across the NA. It is important to note that atmospheric motion is not considered in our product and that the maps resulting from the data represent a static picture of potential sea-level concentrations of MSA and nss-$SO_4^=$, in a certain pixel and at a certain time as a result only of the interplay between local DMS emissions, photochemistry and dilution/removal processes, and that provide accurate predictions of the actual sea level concentrations of MSA and nss-$SO_4^=$ once averaged over 2-to-3-days transport tracts. Accordingly, the IPB-MSA&SO₄ data presented hereafter are different from the output of a chemical transport model. Nevertheless, we believe that this unprecedented dataset may be useful for many research purposes, for instance, investigating long-term trends, or addressing the interannual or spatial variability in the production of biogenic sulfur aerosol species. Examples of the scientific information that can be extracted from the data and on how they can be compared to model output or in-situ observations are provided in the next Sections.

### 4.4 Comparison with CAMS Reanalysis

To further examine the effectiveness of our GPR model, we compared the observed MSA concentrations at MHD with the most recently released CAMS-EAC4 (Inness et al., 2019) reanalysis datasets. The EAC4 (ECMWF Atmospheric

Composition Reanalysis 4) is the fourth generation of the ECMWF global reanalysis dataset of atmospheric composition from the Copernicus Atmosphere Monitoring Service (CAMS). CAMS-EAC4 is a collection of atmospheric composition fields from 2003 to the present, including aerosols and chemical species for which MSA data is available. The spatial resolution of the CAMS datasets is about $0.75° \times 0.75°$ and a 3h temporal resolution. Our datasets have a resolution of $0.25° \times 0.25°$ and start from 1998. To compare the two products, we extracted MSA data from CAMS locally, at the grid cell in front of the MHD station, corresponding to maritime BT timings, and averaged them to daily resolution. Conservatively, the MSA concentration data simulated by GPR were taken from the validation and test sets, which were not included in the model training. Such MSA concentrations at MHD were projected by incorporating predictors along the BTs into consideration to account for the air motion (see Section 3.1.2 for details).

Scatter plots and joint probability histograms of residual errors (Fig. 6) were constructed to compare the accuracy between GPR, CAMS and observations (referred to as OBS). It can be seen from the scatter plots (Fig. 6a and Fig. 6b) that the GPR-simulated MSA best matches the observations, with a 0.84 fitted slope, 0.93 correlation coefficient and most of the data points comprised within the 95% confidence bounds. The joint probability histograms between observed MSA and the residuals (OBS – GPR) and (OBS – CAMS) are used to verify the variance of residual errors around zero. The GPR histograms (Fig. 6c and Fig. 6e) show that the residual errors are mostly centered around zero (dashed black line in the right) up to the value of 0.1 $\mu g\ m^{-3}$ where the majority of data points lie, while CAMS are skewed toward negative residuals followed by positive residuals mainly at high MSA values (Fig. 6d and Fig. 6f). Quantitatively, the GPR has relative MAE equal to 4.3% in comparison to 6.3% for CAMS. In summary, GPR better captures the low concentrations of MSA, which CAMS tends to overestimate, while both CAMS and GPR show limitations in retrieving the extreme points of MSA concentrations. A quantitative statistical analysis (Fig. 6g) showed that no statistically significant ($p<0.05$) difference exists between the seasonal median MSA from OBS and GPR, while CAMS presents a significant ($p<0.05$) difference in all seasons except summer. Nevertheless, the two datasets (GPR and CAMS) properly retrieve the observed MSA seasonal cycle.

### 4.5 Comparison with the Polarstern cruise results

In this Section, we present a case study exemplifying how the IPB-MSA&SO4 datasets can be used. Because the data product represents the concentration of freshly formed sulfur aerosol species and the ML model does not account for atmospheric transport, users must interpret the datasets considering the air mass history. To better clarify the idea, we employed the independent MSA data measured during the Polarstern campaigns in the NA (Huang et al., 2017), which were not used in the training/validation or testing/evaluation of the ML models, and compared them with predicted MSA by GPR. In particular, the MSA by GPR was extracted along air mass BTs arriving at the hourly sites of the ship tracks and then averaged considering a 0-day (simultaneously), 1-day, 2-day and 3-day air mass history. The MSA measurements on Polarstern were performed in four scientific cruises including two spring seasons (April-May 2011 & April-May 2012) and

two autumn seasons (October-November 2011 & October-November 2012). The ship tracks of the cruises from which the data were taken in the present study are shown in Fig. 7. It can be seen that the best match between GPR-simulated MSA and observed MSA occurred when 2-day air masses were considered. At 2-day air mass history, the slope reached 0.78 and the correlation coefficient 0.81 (Fig. 7a-d). Again, as seen in Fig. 7f, GPR MSA is considerably more consistent with observations than CAMS, for which a significant difference with observations ($p < 0.05$) can be appreciated.

## 4.6 Spatial distributions of MSA and nss-SO$_4^=$

In order to elucidate the geographical distributions of biogenic sulfur aerosol production across the NA domain, the IPB-MSA&SO$_4$ datasets in the 25 years (1998–2022) were averaged to obtain the climatic annual and monthly distributions of MSA and nss-SO$_4^=$ as illustrated in Fig. 8 and Fig. 9. Across the NA domain, the annual average of MSA is $0.016 \pm 0.007$ µg m$^{-3}$, whereas the annual average of nss-SO$_4^=$ is $0.250 \pm 0.077$ µg m$^{-3}$ (Table S2). The annual spatial distributions of MSA and nss-SO$_4^=$ exhibit a latitudinal gradient over the majority of the NA area that increases from north to south, except below nearly 35° N, where it increases from west to east. Notwithstanding, the latitudinal gradients are much more evident than longitudinal variations. For instance, MSA grows at a rate of 0.0016 ($R^2 = 0.93$; $p<0.05$) µg m$^{-3}$ per each 1° latitude towards the south and 0.00036 ($R^2 = 0.53$; $p<0.05$) µg m$^{-3}$ per each 1° longitude eastward (it reaches its peak between 20° and 10° W). Furthermore, for each 1° southward, nss-SO$_4^=$ increases by 0.0212 ($R^2 = 0.96$; $p<0.05$) µg m$^{-3}$, whereas there are no significant changes in nss-SO$_4^=$ with longitude ($R^2 = 0.01$; $p>0.05$). The highest concentrations of both components (>90[th] percentile) are primarily found in the southeast of the domain (in front of the Moroccan coast and the Gibraltar strait). Minimum annual concentrations (<10[th] percentile) are found in northern areas of the domain, particularly in the Labrador Sea and near the shores of Greenland and Iceland.

The annual average MSA to nss-SO$_4^=$ (MSA:nss-SO$_4^=$) ratio is $0.053 \pm 0.012$ (Table S2), with a consistent latitudinal gradient increasing southward (rate of change = 0.0028 ($R^2 = 0.93$; $p<0.05$) per each 1° latitude). The lower MSA: nss-SO$_4^=$ values are found in the northwest of the domain, while the higher values are apparent in front of the African coast, the ratio is practically constant across the same latitudinal band. It is worth evidencing that the region with extremely high MSA concentrations and high MSA:nss-SO$_4^=$ (above the mean + three times the standard deviations) is linked to the Canary upwelling system on the northwest African coast. The Canary Current system is one of the world's most productive regions of the ocean, known as eastern boundary upwelling systems (EBUSs) (Chavez and Messié, 2009; Carr, 2001). This may indicate a link between EBUSs and the potential formation of biogenic aerosol concentrations in the atmosphere. Previous research has shown how EBUSs changed in response to climate change (Bograd et al., 2023; Sydeman et al., 2014; Bonino et al., 2019), including the trend toward increased upwelling intensity (Wang et al., 2015; García-Reyes et al., 2015);

however, little is known about the impact of EBUSs on marine biogenic emissions and the resultant aerosol fluxes. Future studies are needed to address these issues in order to better understand the role of EBUSs on the aerosol-climate systems.

Looking at the monthly climatological maps (Fig. 9), it is revealed that MSA and nss-SO$_4^=$ display a gradual increase in their concentrations southward, clearly evident from October to March, resulting in a large difference between the northern and southern parts of the domain. On the contrary, during summer, the concentrations are more homogeneous over the domain (see latitudinal patterns in Fig. 9), still with a tendency to higher concentrations over the northeastern part. The seasonality of MSA and nss-SO$_4^=$ is evident: the increase for both compounds starts in April and peaks in June-July followed by a gradual decrease in September (Fig. S9 and Table S2). The lowest MSA (nss-SO$_4^=$) concentration occurs in December at 0.006 ± 0.005 (0.155 ± 0.079) µg m$^{-3}$ and the highest occurs in June at 0.029 ± 0.013 (0.364 ± 0.075) µg m$^{-3}$ (Table S2), consistent with the fact that winter and summer are typically the lowest and highest seasons for biological activity, respectively for the NA (Mansour et al., 2023a). The coefficient of variation (COV), defined as the ratio between the standard deviation and the mean value, expressed in percentage, at each grid point, is used to assess how much the MSA and nss-SO$_4^=$ vary around their mean value in each month; variability increases with higher COV. The maps (Fig. S10) confirm that the variability of sulfur aerosol species depends strongly on the season; MSA and nss-SO$_4^=$ are mostly stable (little variations) during the winter, whereas most variations occur between April (late spring) and June (early summer) and preferentially over the eastern part of the NA than the western.

The MSA:nss-SO$_4^=$ also exhibits a seasonal pattern, with the lowest (highest) values observed during the winter (summer), as presented in Fig. 9c. July has the highest spatial average of the ratio of 0.077 ± 0.022 while the lowest of 0.032 ± 0.012 occurs in December (Table S2). Looking at the overall distributions, MSA:nss-SO$_4^=$ demonstrates a general southward increase, with the exception of summer months. In summer (mainly July and August), MSA:nss-SO$_4^=$ above 50°N has an opposite trend with respect to the one below 50°N. In detail, from North to South, we report a sharp increase in MSA:nss-SO$_4^=$, maximized around 50°N, followed by an abrupt decrease toward the equator. The possible explanation for the decline in MSA:nss-SO$_4^=$ below 50°N is that the reduction in MSA:nss-SO$_4^=$ is related to an increase in AT caused by warmer air nearing the equator, in line with observations in the Pacific Ocean (Bates et al., 1992) and with the higher ratio observed in colder air masses (marine Polar and Arctic) with respect to warmer ones (marine Tropical) at MHD (Ovadnevaite et al., 2014). As a final remark, we report that the summertime low MSA:nss-SO$_4^=$ below 50°N is linked to a decrease in F$_{DMS}$ in the same latitudinal zone (Mansour et al., 2023a). Owing to the low DMS emissions, the different DMS oxidation patterns may be in competition (Barone et al., 1995); since MSA is formed preferentially through the pathway of OH addition at low temperatures (Shen et al., 2022), the production of MSA may be decreased relative to that of nss-SO$_4^=$ in the warm southern part of the domain, during summer, leading to the observed decrease in the MSA:nss-SO$_4^=$ ratio.

## 5. Data availability

The dataset includes daily MSA and nss-$SO_4^=$ concentrations at 0.25° × 0.25° spatial resolution over the North Atlantic Ocean from January 1998 to December 2022. The datasets are publicly available in NetCDF format as daily files on the Mendeley online repository at https://doi.org/10.17632/j8bzd5dvpx.1 (Mansour et al., 2023b).

550 ## 6. Conclusions

Marine aerosol data can be obtained from in-situ coastal observatories or from shipborne measurements, however, punctual coast observations are limited under the point of view of the spatial representativity, while shipborne measurements suffer of limitations in terms of temporal coverage. Understanding the dynamics of marine-derived biogenic sulfur aerosols and their radiative effects, as well as carrying out relevant scientific studies, requires long-term, continuous and high-resolution (space 555 and time-wise) datasets. To overcome the limitations of punctual measurements, we combined the in-situ observations of sulfur aerosol data at Mace Head and from NAAMES cruises, as dependent variables, and the sea-to-air DMS flux and ECMWF-ERA5 reanalysis meteorological datasets, as independent variables, to investigate the potential of machine learning techniques for the prediction of daily MSA and nss-$SO_4^=$ sea-level concentrations over the North Atlantic Ocean. We evaluated five machine learning models (*i.e.,* SVM, DT, RE, GPR, and ANN), considering various sets of hyperparameter 560 optimizations. Our findings demonstrated that the GPR model outperforms other approaches in simulating the concentrations of biogenic sulfur aerosols, capturing up to 86% and 72% of the observed variance in daily MSA and nss-$SO_4^=$, respectively. This makes the GPR an effective tool for obtaining trustworthy sea-level MSA and nss-$SO_4^=$ concentrations over the North Atlantic, which may also be successful in other oceanic regions or over the entire global ocean. The impact of the six independent predictors on the simulated MSA and nss-$SO_4^=$ is further evaluated using the GPR partial dependence analysis, 565 which reveals that the relationships between them are multifaceted rather than linear or monotonically varying.

By the GPR machine learning method, we constructed a novel 0.25°×0.25° resolution daily gridded dataset of in-situ produced biogenic MSA and nss-$SO_4^=$ concentrations (named IPB-MSA&SO$_4$) covering the North Atlantic Ocean from 1998 to 2022. The dataset represents the sea-level concentrations of MSA and nss-$SO_4^=$ associated with in-situ production in the MBL, i.e., the concentration of sulfur aerosol species resulting, in each pixel, from the local biogenic emissions in 570 combination with local atmospheric conditions. Other inputs, such as terrestrial emissions or sinking of sulfur species produced in the free troposphere are not accounted for in the present dataset.

Comparison of the GPR-derived MSA with existing CAMS-EAC4 reanalysis product reveals that our high-resolution dataset accurately reproduces the spatial and temporal patterns of the biogenic sulfur aerosol concentration and has high consistency with the independent observations of the Polarstern cruises measurements in the Atlantic. The obtained IPB-MSA&SO$_4$ data 575 were used to analyze the spatiotemporal variations of MSA, nss-$SO_4^=$, and the ratio between them (MSA:nss-$SO_4^=$). It was

found that the monthly concentrations of MSA and nss-$SO_4^=$ across the NA are characterized by a significant southward increase in each month, with the exception of summertime when MSA and nss-$SO_4^=$ displayed more homogeneous spatial patterns with a tendency to higher concentrations over the northeastern part of the domain. The MSA:nss-$SO_4^=$ ratio exhibits a seasonal variation from winter (low) to summer (high) characterized by a sharp decline from the 50 °N parallel toward the equator mainly in July-August. In general, the atmospheric concentration of sulfur aerosol species tends to be more stable in winter, whereas wider variations are associated with late spring and early summertime and more with the eastern part of the domain than to the western one.

More in-depth analyses can be conducted based on the presented biogenic sulfur aerosol concentration dataset, which could help further understanding of oceanic sulfur-aerosol-cloud interactions. For instance, we evidence that the Canary eastern upwelling system emerges from the dataset as a hotspot of high sea-level MSA concentration and high MSA:nss-$SO_4^=$ ratio; such a finding is worth further investigation and may shed light on the role of EBUSs in the production of biogenic marine aerosols and on its climate relevance.

**Author contributions**

KM and MR contributed to the conceptualization and design of the study. KM organized the datasets, constructed the models, analysed the data, and visualized the results. KM wrote the first draft of the manuscript under the supervision of MR. KM, MR, SD, DC, JO, LMR, MP, LP, SH, and CO contributed to the results investigation, manuscript revision, reading and editing, and approved the submitted version.

**Competing interests**

The authors declare that the research was conducted in the absence of any commercial or financial relationships that could be construed as a potential conflict of interest.

**Acknowledgements**

We gratefully acknowledge the Copernicus climate change service (C3S) for the provision of ECMWF-ERA5 reanalysis meteorological data and the NOAA Air Resources Laboratory (ARL) for the provision of the HYSPLIT transport and dispersion model. University of Galway team acknowledges the support from Irish EPA (AC3 and AEROSOURCE, 2016-CCRP-MS-31) and the Department of Environment, Climate and Communications as well as MaREI, the SFI Research Centre for Energy, Climate, and Marine.

**Financial support**

The research was funded by the European Commission, H2020 Research Infrastructures, project FORCeS (grant no. 821205).

Figure 1: The study region of the North Atlantic Ocean (72° – 0° W, 20° – 66° N) with bathymetry presented in meters. The gridded bathymetric dataset was extracted from the General Bathymetric Chart of the Oceans (https://www.gebco.net), the GEBCO_2023 Grid. The green-filled pentagram represents the Mace Head measuring station on the west coast of Ireland and the dark-red points are the sampling points that represent marine conditions in the NAAMES cruises track. The violet
points represent the ship track during Polarstern campaigns.

Figure 2: The methodology's workflow. Predictors and response variables data preparation, the overall framework of generation and development of the trained models, including a schematic diagram of 5-fold cross-validation, models export and validation details, as well as post-processing analysis.

Figure 3: Comparison of predicted and observed MSA on the hourly (left panels) and daily (right panels) scales: (a) GPR, (b) EBT, (c) SVM, (d) ANN, and (e) DT. The validation and test data subsets are used to compute the model's performance. $R^2$ and RMSE are computed in a logarithmic space, whereas MAE is computed on a normal scale.

Figure 4: Comparison of predicted and observed nss-$SO_4^=$ on the hourly (left panels) and daily (right panels) scales: (a) GPR, (b) EBT, (c) SVM, (d) ANN, and (e) DT. The validation and test data subsets are used to compute the model's performance. $R^2$ and RMSE are computed in a logarithmic space, whereas MAE is computed on a normal scale.

Figure 5: Partial dependence plots of MSA and nss-$SO_4^=$ as a function of the predictors revealed by the GPR model.

Figure 6: Comparison between observed MSA at MHD measuring site and both MSA predicted by GPR (a) and MSA extracted from CAMS reanalysis (b). (c) and (d): joint probability histograms between observed MSA and residual errors (observed–predicted); the black dashed lines represent the change of MSA residual errors in each bin. MAE is the mean absolute error, and the relative MAE has been calculated as the MAE divided by the range of observed MSA. (e) and (f):
frequency distributions of the residual errors. (g): Seasonal box charts from different datasets. Each box chart displays the

median (line inside of each box), the 1$^{st}$ and 3$^{rd}$ quartiles (bottom and top edges of each box), the minimum and maximum values that are not outliers (whiskers), and any outliers represented by '+' (computed as values that are more than 1.5 of the interquartile range away from the top or bottom of the box). Box charts whose notches (the shaded region around each median) do not overlap have different medians at the 95% confidence level.

Figure 7: (a) Scatter plots between observed MSA during the Polarstern campaigns (Huang et al., 2017) and predicted MSA by GPR, considering (a) 0-day, (b) 1-day, (c) 2-day and (d) 3-day air mass history. (f): Seasonal box charts from different datasets. The features displayed on each box chart are the same as those given in Fig. 6.

Figure 8: The annual averages of (a) MSA, (b) nss-$SO_4^=$, and (c) MSA:nss-$SO_4^=$ spatial distributions based on GPR at 0.25° × 0.25° resolution during the 1998–2022 period. The latitudinal (longitudinal) gradients of each component are displayed in the left (bottom) panels whereas shaded areas represent ± standard deviations. The black crosses evidence the extremely high concentrations, more than three times standard deviations plus the annual mean climatology.

Figure 9: Monthly spatial distributions of (a) MSA, (b) nss-$SO_4^=$, and (c) MSA:nss-$SO_4^=$ based on GPR over 1998–2022 at 0.25° × 0.25° resolution. (d) Monthly latitudinal distributions of each component while shaded areas represent ± standard deviations.

Table 1: List of machine learning models used in the present study.

Table 2: The Pearson's Coefficients between possible predictors at the selected marine air masses and the in-situ observed MSA and nss-$SO_4^=$ concentrations. The MSA, nss-$SO_4^=$ and $F_{DMS}$ values are used in the log scale. All values are statistically significant at $p<0.05$. Bolds evidence the maximum during different days of air mass history.

Table 3: Details of the number of hourly (MSA and nss-$SO_4^=$) data points corresponding to selected marine BTs. The threshold used for filtering outlier values, and the number of data points after filtering are given.

Table 4: Multilinear regression of MSA and nss-$SO_4^=$ as a function of predictors. The MSA, nss-$SO_4^=$ and $F_{DMS}$ values are used in the log scale. Each independent variable's contribution to $R^2$ is the decrease in total $R^2$ when that variable is eliminated. Individual $R^2$ contributions are normalized and added together to equal the overall $R^2$. According to the analysis of variance (ANOVA) on the multilinear regression models, all predictors contribute statistically significantly ($p<0.05$) to the MSA and nss-$SO_4^=$ variance.

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

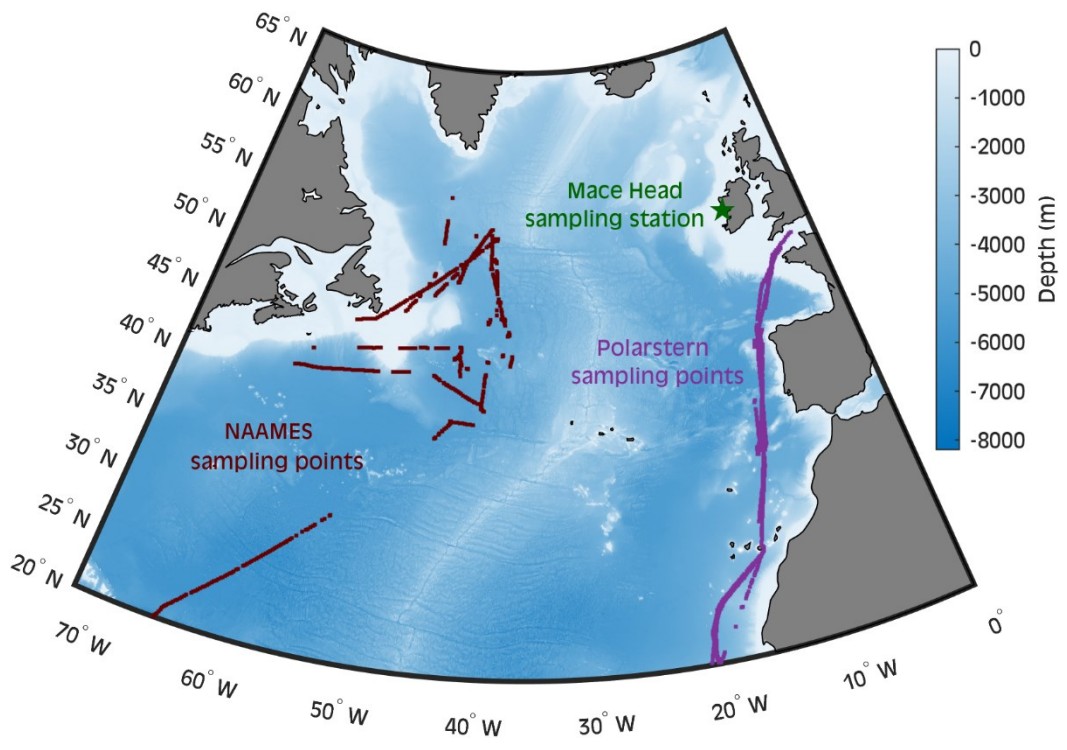

**Fig. 1**

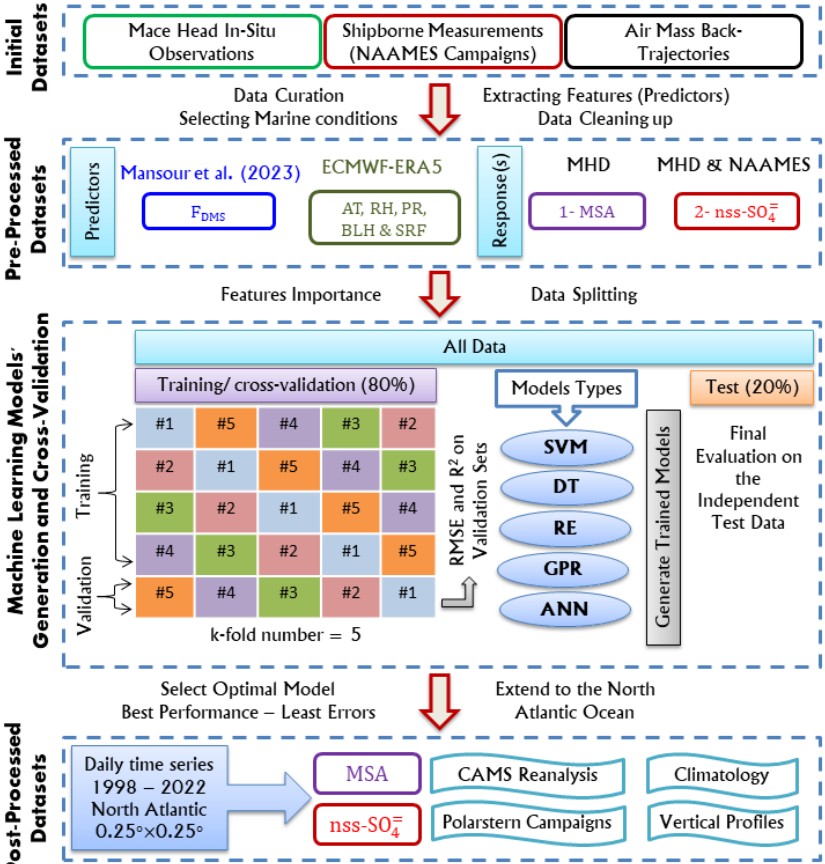

**Fig. 2**

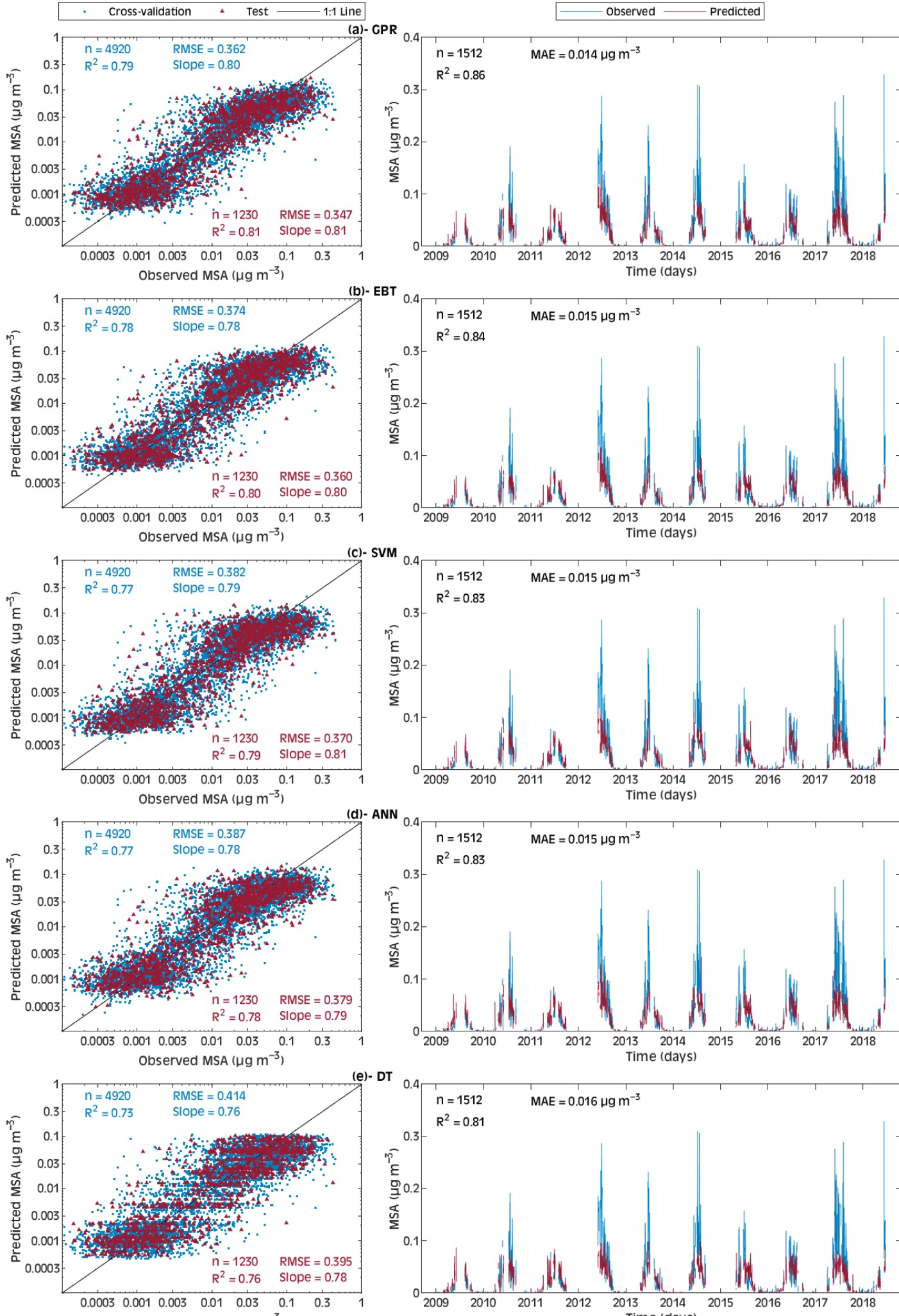

**Fig. 3**

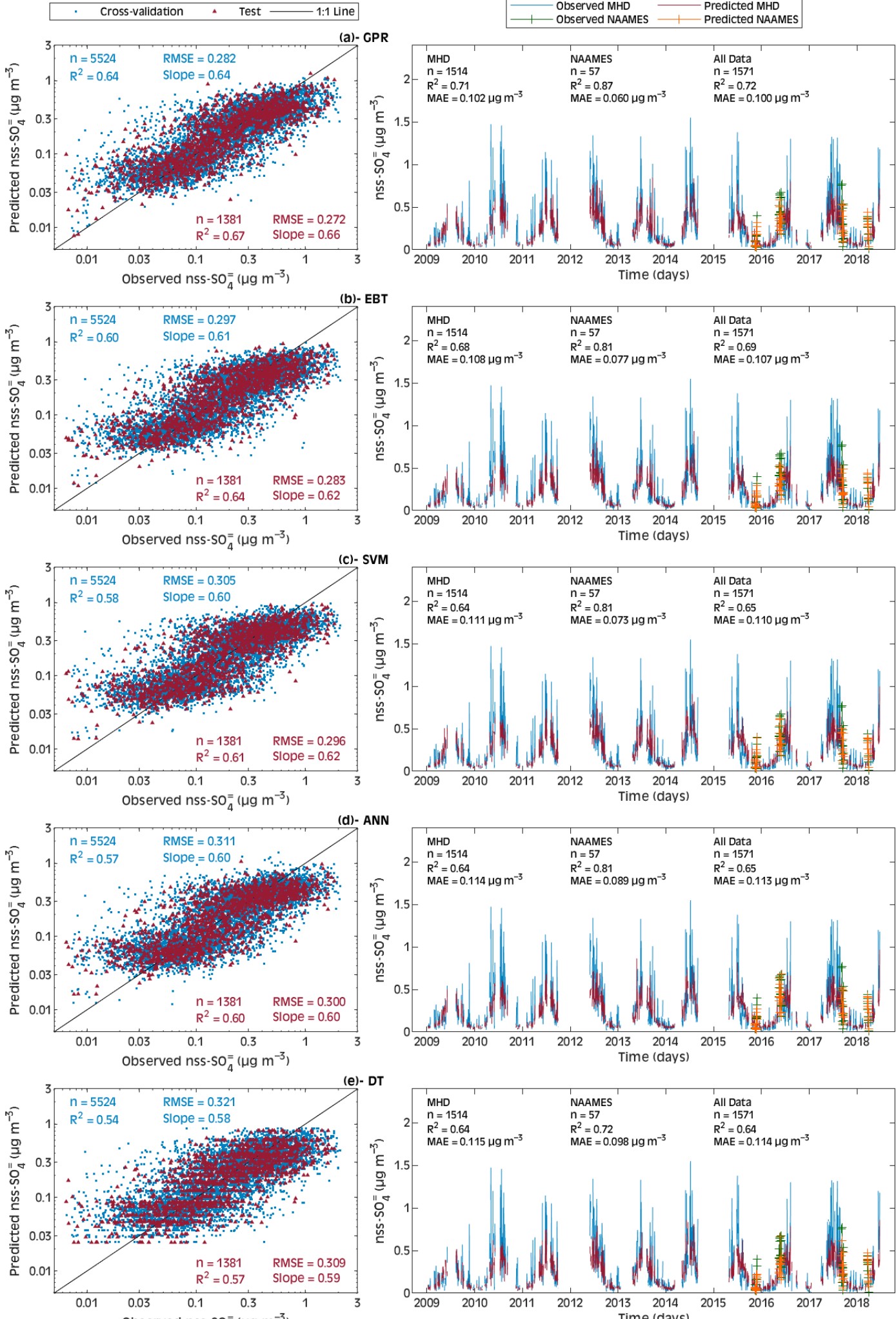

**Fig. 4**

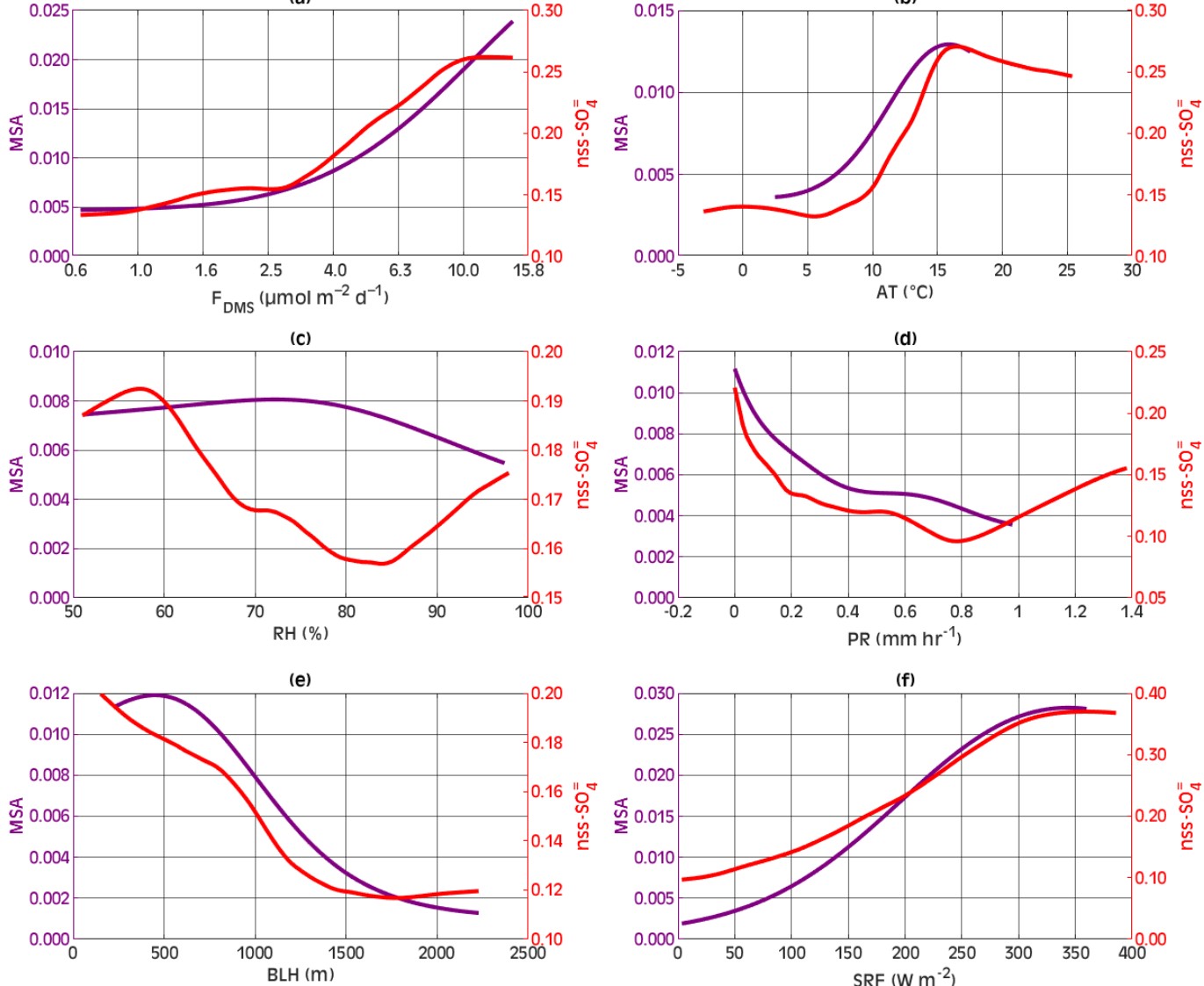

Fig. 5

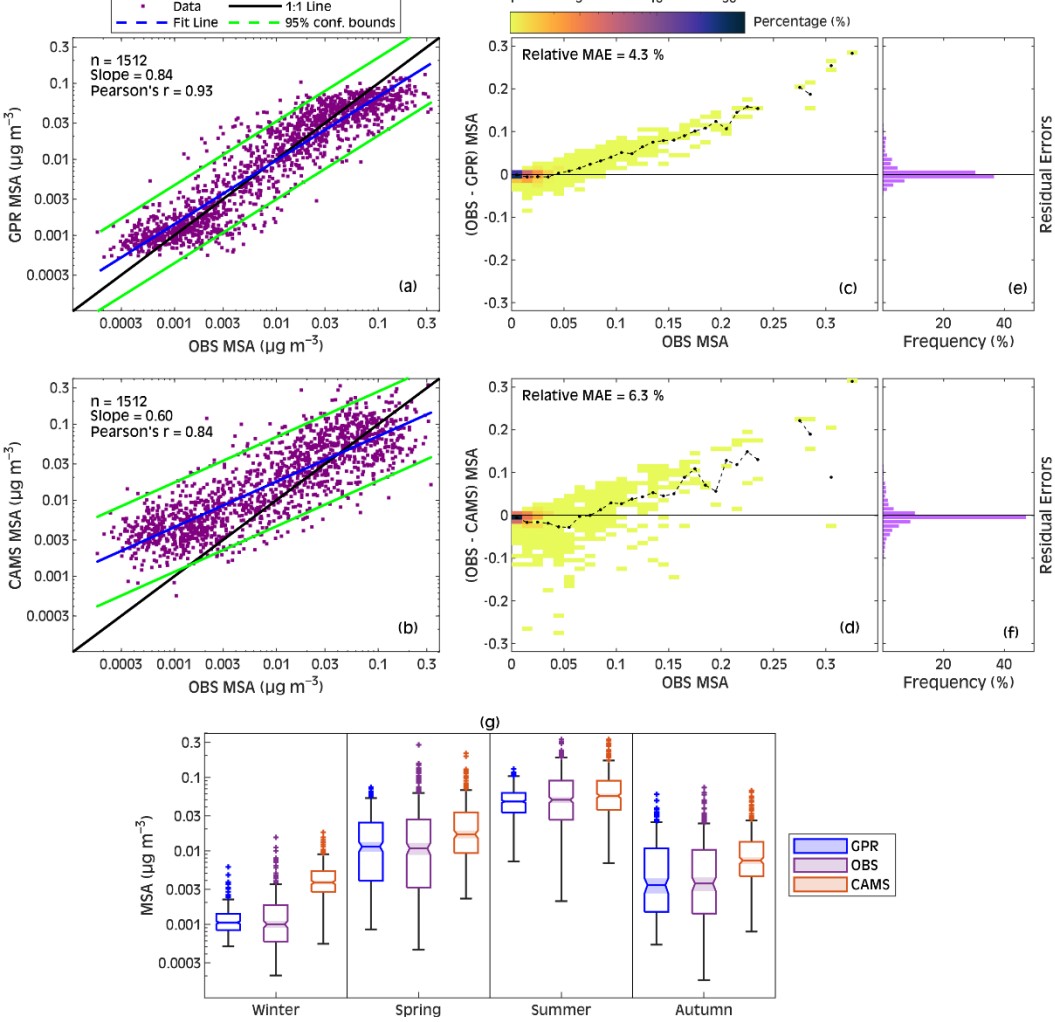

**Fig. 6**

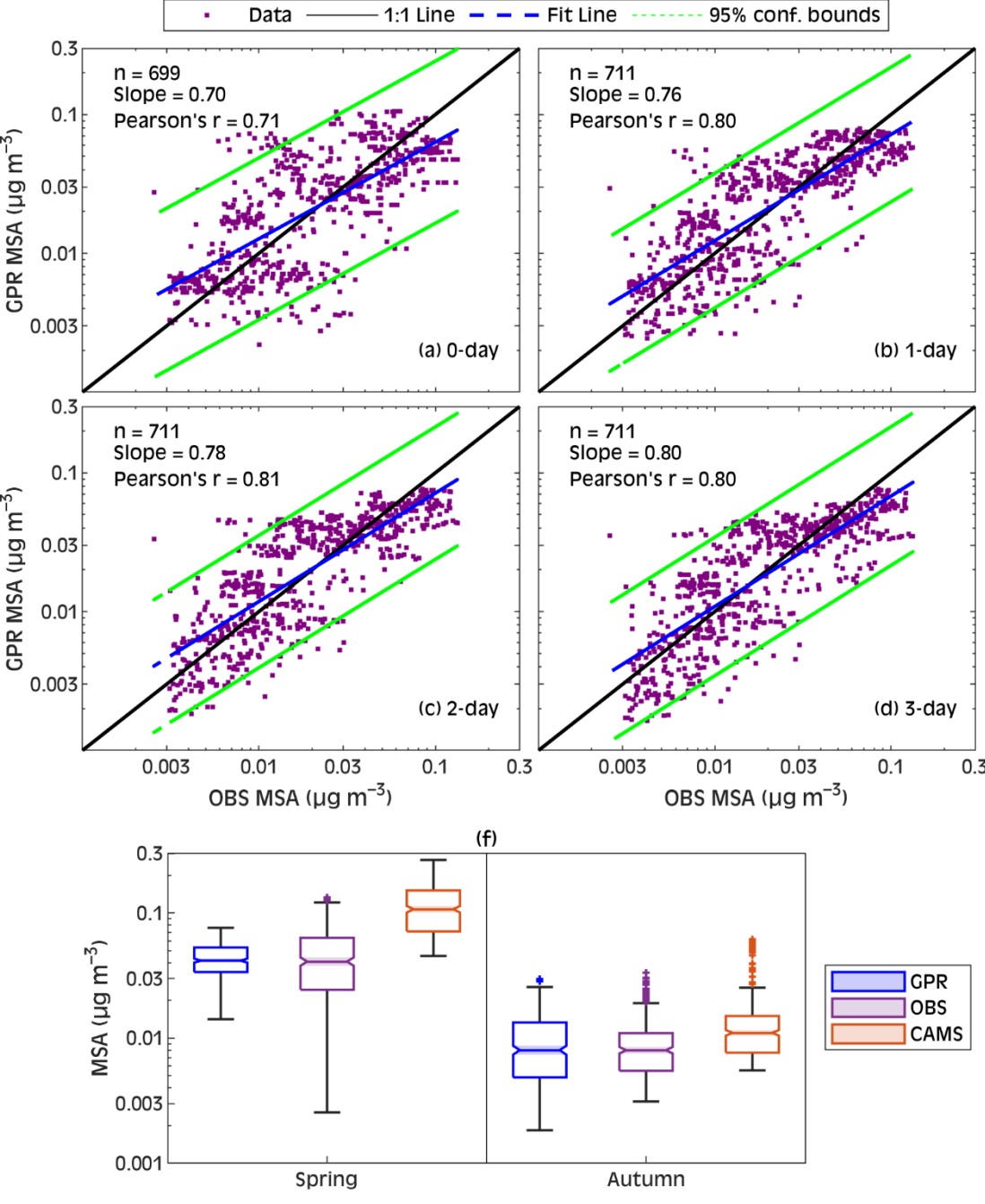

**Fig. 7**

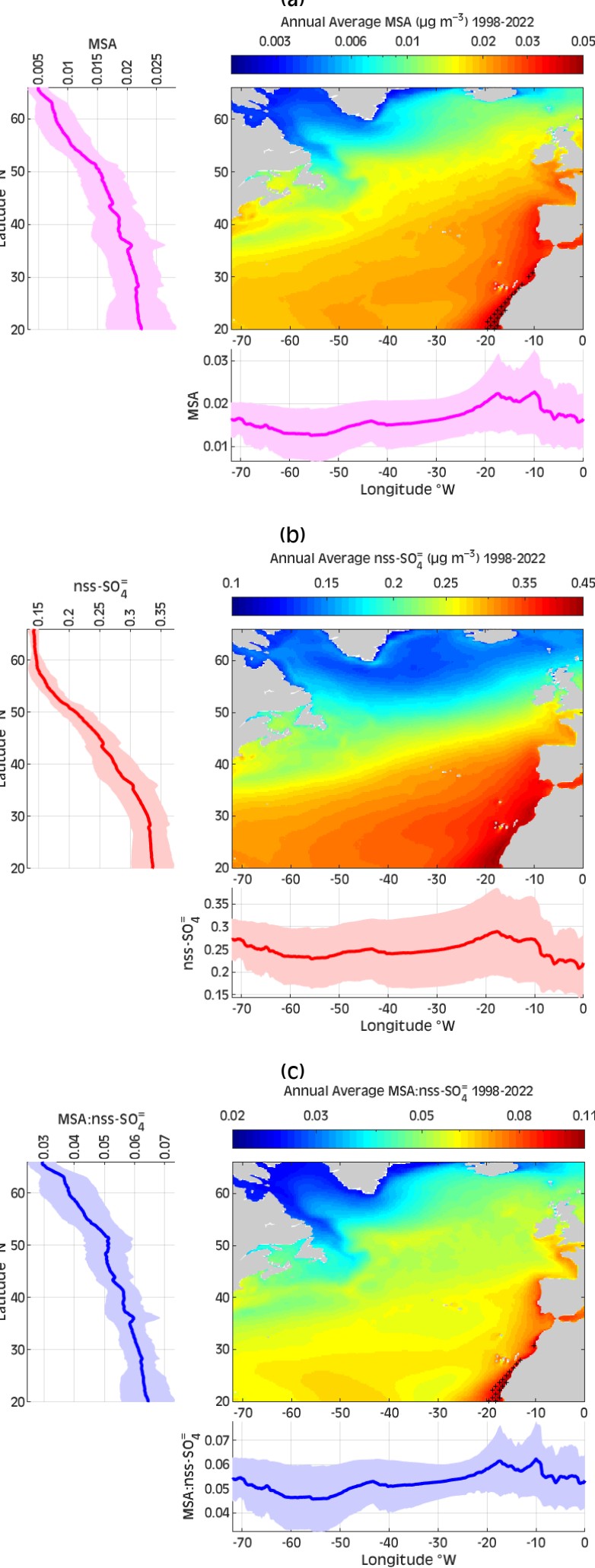

**Fig. 8**

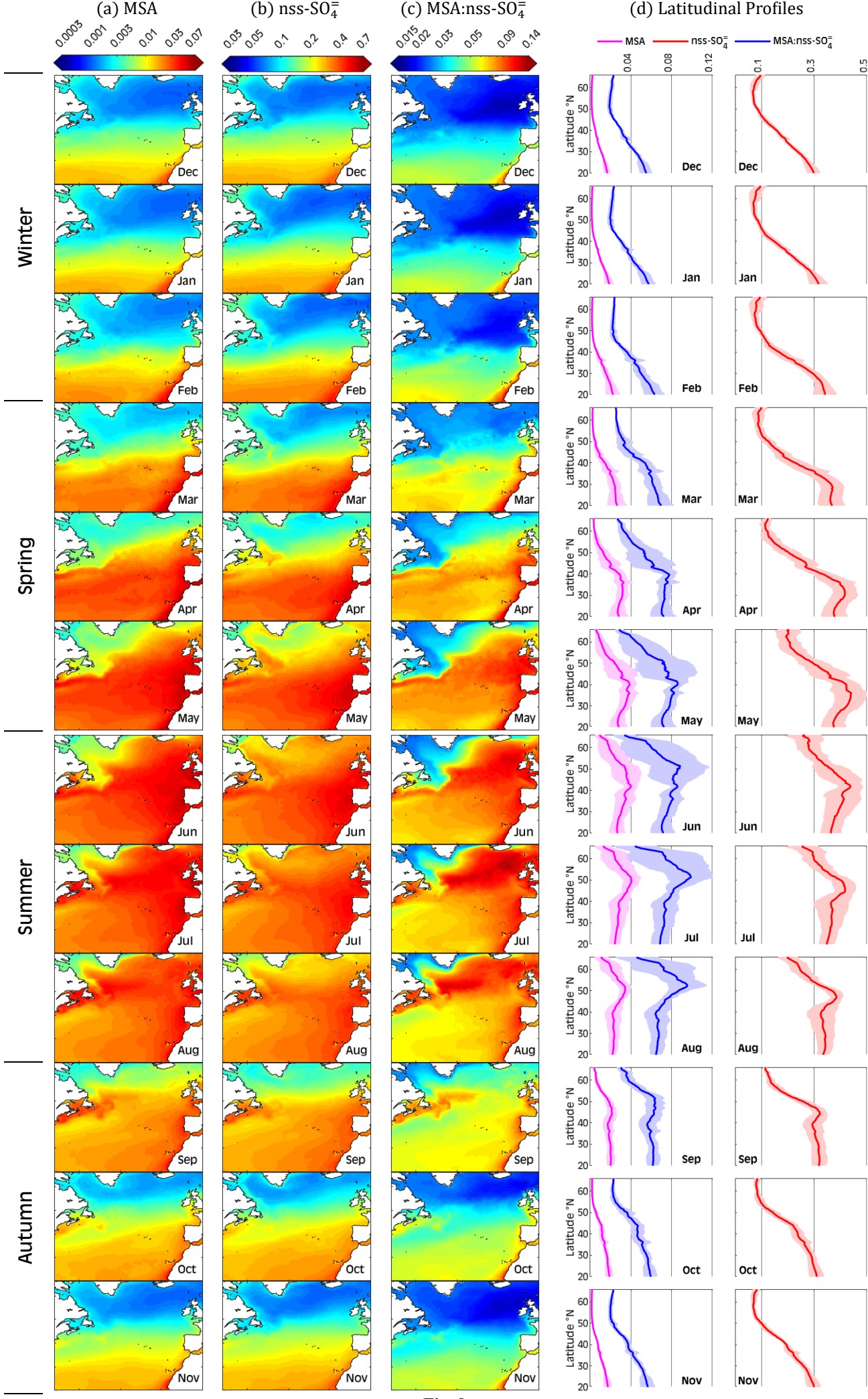

**Fig. 9**

| Model Type | Preset | Hyperparameters if any |
|---|---|---|
| **Support Vector Machines** | Linear | |
| | Quadratic | |
| | Cubic | |
| | Fine Gaussian | Kernel scale = 0.61 |
| | Medium Gaussian | Kernel scale = 2.4 |
| | Coarse Gaussian | Kernel scale = 9.8 |
| **Decision Tree** | Fine | Minimum leaf size = 4 |
| | Medium | Minimum leaf size = 12 |
| | Coarse | Minimum leaf size = 36 |
| **Regression Ensemble** | Boosted | Minimum leaf size = 8 Number of learners = 30 |
| | Bagged | Minimum leaf size = 8 Number of learners = 30 |
| **Gaussian Process Regression** | Squared Exponential | |
| | Matern 5/2 | |
| | Exponential | |
| | Rational Quadratic | |
| **Neural Networks** | Narrow | Number of fully connected layers = 1 First layer size = 10 |
| | Medium | Number of fully connected layers = 1 First layer size = 25 |
| | Wide | Number of fully connected layers = 1 $1^{st}$ layer size = 100 |
| | Bi-layered | Number of fully connected layers = 2 $1^{st}$ layer size = 10 $2^{nd}$ layer size = 10 |
| | Tri-layered | Number of fully connected layers = 3 $1^{st}$ layer size = 10 $2^{nd}$ layer size = 10 $3^{rd}$ layer size = 10 |

**Table 1**

| Predictors | BT length (days) | MSA | nss-$SO_4^=$ | |
| --- | --- | --- | --- | --- |
| | | MHD | MHD | NAAMES |
| $F_{DMS}$ | 0 | 0.27 | 0.24 | 0.04* |
| | 1 | 0.64 | 0.53 | 0.24 |
| | 2 | 0.66 | 0.54 | 0.38 |
| | 3 | **0.69** | **0.55** | **0.47** |
| AT | 0 | **0.65** | **0.61** | 0.17 |
| | 1 | 0.57 | 0.56 | 0.29 |
| | 2 | 0.53 | 0.53 | 0.35 |
| | 3 | 0.53 | 0.51 | **0.37** |
| RH | 0 | 0.15 | 0.15 | 0.27 |
| | 1 | 0.33 | 0.27 | 0.22 |
| | 2 | 0.39 | 0.31 | 0.24 |
| | 3 | **0.44** | **0.33** | **0.28** |
| PR | 0 | -0.18 | -0.12 | -0.09 |
| | 1 | -0.27 | -0.26 | -0.27 |
| | 2 | -0.33 | -0.31 | **-0.34** |
| | 3 | **-0.35** | **-0.33** | -0.32 |
| BLH | 0 | -0.41 | -0.32 | -0.32 |
| | 1 | -0.53 | -0.45 | -0.34 |
| | 2 | -0.58 | -0.49 | **-0.36** |
| | 3 | **-0.60** | **-0.49** | -0.35 |
| SRF | 0 | 0.32 | 0.23 | 0.14 |
| | 1 | 0.73 | 0.61 | 0.53 |
| | 2 | 0.77 | 0.65 | 0.62 |
| | 3 | **0.78** | **0.67** | **0.63** |

**Table 2**

| Response | No. of hourly data points | Lower threshold according to 0.1 percentile | Upper threshold according to 99.9 percentile | No. of data lost due to filtering | No. of hourly data points after cleanup |
|---|---|---|---|---|---|
| MSA | Source: MHD $n = 6162$ | 0.0001 $\mu g\ m^{-3}$ | 0.45 $\mu g\ m^{-3}$ | 12 | 6150 |
| nss-SO$_4^=$ | Source: MHD $n = 6260$ | 0.006 $\mu g\ m^{-3}$ | 2.116 $\mu g\ m^{-3}$ | 12 | 6905 |
| | Source: NAAMES $n = 660$ | 0.007 $\mu g\ m^{-3}$ | 1.107 $\mu g\ m^{-3}$ | 3 | |

**Table 3**

| | Total explained variance by $R^2$ | Normalized Contribution to $R^2$ (%) | | | | | |
|---|---|---|---|---|---|---|---|
| | | AT | RH | PR | BLH | SRF | $F_{DMS}$ |
| MSA | 74.36% | 6.86 | 0.47 | 2.82 | 8.66 | 42.77 | 12.97 |
| nss-SO$_4^=$ | 53.39% | 11.64 | 0.55 | 5.07 | 3.63 | 25.83 | 6.66 |

**Table 4**