# Peer review of "IPB-MSA&SO4: a daily 0.25° resolution dataset of In-situ Produced Biogenic Methanesulfonic Acid and Sulfate over the North Atlantic during 1998–2022 based on machine learning"

_Earth System Science Data, 2023_

## Author Comment (AC1)

**Answers to Referee 1**

We would like to express our gratitude to the reviewer for their insightful comments on the manuscript. According to the comments, the manuscript has been revised. Please find our replies in the pdf file. In the responses, the reviewer's comments are in black text, and our responses are in blue and *the main text modifications to the revised manuscript are in italics*.

Mansour et al. used the machine-learning model to predict the biogenic methanesulfonic acid (MSA) and sulfate (SO4) concentrations covering the North Atlantic Ocean. Overall, the study is very interesting and falls into the scope of ESSD. However, the manuscript still suffers from some major weaknesses. I recommend the manuscript for publication on ESSD after the following comments have been well addressed.

1. The novelty of this dataset in this study should be well clarified in the introduction.

The introduction has been updated as follows:

*"In this study, we present the first high-resolution and long-term daily gridded time series of freshly formed In-situ Produced Biogenic Methanesulfonic Acid and Sulfate (IPB-MSA&SO4) concentrations over the NA ocean at 0.25° × 0.25° spatial resolution. The data covers 25 years from 1998 to 2022 with the possibility of future updating year by year. The dataset is a unique and novel product that in fact extends the space and time representativeness of atmospheric in-situ observations of marine aerosol chemical properties over the North Atlantic Ocean, by exploiting the potential of machine learning. The dataset indeed represents the sea-level concentrations of MSA and SO4, in each grid point of the domain, resulting from the interplay between precursor emissions and local atmospheric conditions."*

In addition, the following sentence has been added as a complement to the 2ⁿᵈ paragraph:

*"Recently, multilinear regression was utilized to simulate monthly MSA over the eastern China seas at a spatial resolution of 1° × 1° (Zhou et al., 2023), concluding that MSA spatial/seasonal patterns exhibit significant variability, which is primarily governed by surface phytoplankton biomass and the atmospheric boundary layer height."*

2. Why do not you use the physical model (e.g., MITgcm, ROMS) output as the input for the machine-learning model? As shown in figure 1, the middle region of Atlantic lacks of measurement, and this region might show large uncertainties based on machine-learning models alone.

Here it is necessary to clarify that the only data missing in the central part of the domain (Fig. 1) are the atmospheric observations of MSA and nss-SO$_4^=$ concentrations. These cannot be accessed from the output of the physical models suggested by the reviewer. Furthermore, the aim of this work is to provide an alternative to uncertain model outputs of MSA and SO$_4^=$ concentrations, by using an original approach to extend the time and space representativeness of in-situ measurements. All the other data used to generate the IPB-MSA&SO$_4$ dataset come from satellite observations or previously generated datasets which cover the whole domain. For instance, one of the main predictors used for calculating MSA and SO$_4^=$ atmospheric concentrations, is the sea-to-air DMS flux (F$_{DMS}$), serving as the main tracer of marine biogenic sulfur aerosol concentrations in the atmosphere. The F$_{DMS}$ data product has been parametrized from seawater DMS concentrations and wind speed (Mansour et al., 2023). Ultimately, F$_{DMS}$ is a daily (0.25° × 0.25°)– Level 4 product, covering the whole North Atlantic domain, which means that the considered domain has no missing data points. The DMS has been reconstructed by merging high-resolution satellite data (chlorophyll-a concentration, sea surface temperature, and photosynthetically active radiation), Ocean Physics Reanalysis (oceanic mixed layer depth and seawater salinity) and Ocean Biogeochemistry Hindcast (sea surface concentrations of nitrate and silicate).

As the reviewer, we were also worried that the scarcity of the MSA and SO$_4^=$ observational data in the central part of the domain may have resulted in biased (less constrained) predictions in such part of the domain. As a confirmation of the validity of the implemented GPR model, we evaluated the possibility of reconstructing daily variations of nss-SO$_4^=$ during NAAMES campaigns (in the westernmost part of the study area) in the worst-case scenario of training the GPR model only using the MHD data (i.e., measurements collected at the Easternmost side of the domain). This exercise provides an idea of the reliability of the ML approach to model MSA and SO$_4^=$ concentrations in regions poorly constrained by in-situ observations. The results are shown in the following figure displaying that GPR, also in this worst-case scenario deployment, can explain 55% of the daily observed nss-SO$_4^=$ variance. It is also worth considering that the dataset used for this test is limited (*n=57, as the days of observations available from NAAMES*) and does not cover a full seasonal cycle , which makes it harder for the model to perform a good prediction as the seasonality is the main

driver of the variability in biogenic aerosol emissions in the studied domain. Anyhow, we consider the performance of GPR in this worst-case scenario test as more than acceptable and believe consequently that the IPB-MSA&SO4 dataset may be considered reliable also in the central part of the NA, where measurements of MSA and nss-SO$_4^=$ are missing.

In the revised manuscript, as a complement to Section 4.1 "Evaluation of ML model performance", we added the following paragraph also adding the Figure to the Supplement materials.

*"Knowing that the GPR model could be biased due to the inhomogeneous distribution of in situ observations, we assessed the applicability of the GPR model in regions poorly covered by atmospheric observational data (as the central part of the domain) by running the model in a worst-case scenario deployment. In this exercise, we predicted the daily variations of nss-SO$_4^=$ measurements in the westernmost portion of the study area by training the model only with observations from the eastern part of the domain (i.e., data collected at MHD). In this case, MHD data were used for training/cross-validation, while the four NAAMES campaigns were employed as independent test data. The evaluation on the test data (Fig. S8) reveals that GPR can explain 55% of the daily observed nss-SO$_4^=$ variance (MAE= 0.129 µg m$^{-3}$), even in this worst-case scenario and on a limited test dataset (n=57). This more than acceptable performance of the model supports the reliability of the IPB-MSA&SO$_4$ dataset also in the central part of the NA, where measurements of MSA and nss-SO$_4^=$ are missing. In addition, Section 4.5 describes the validation of the GPR model for predicting observed MSA concentrations during the Polarstern campaigns, which were not included in either the model training/cross-validation or in the model test."*

[Figure]

*Figure S8 (revised supplementary): Comparison between daily observed and GPR-predicted nss-SO$_4^=$ during the four NAAMES campaigns. The GPR was trained on the MHD data and tested on the NAAMES data. $R^2$ is computed in a logarithmic space, whereas MAE is computed on a normal scale.*

3. You used many variables to train the machine-learning model. However, I felt these predictors were not strong proxy for MSA and sulfate. Why do not use SO2 satellite product for sulfate estimation?

We respectfully disagree with the reviewer's point of view. SO$_2$ in the atmosphere comes from both natural (e.g., volcanic activity) and anthropogenic sources (industrial processes, fossil fuel combustion by power plants, ships, and other vehicles) with anthropogenic sources contributing importantly at the global level. In this work, we are interested in predicting marine natural biogenic sulfur aerosol concentrations, hence we limit the predictors to variables that can be used as unambiguous tracers of biogenic marine emissions, such as the DMS flux. DMS is the main precursor of MSA and nss-SO$_4^=$ in the marine boundary layer according to at least 40 years of literature, while SO$_2$ observations may be biased by anthropogenic or volcanic inputs.

The goodness of our choice of predictors is proved by our results: the machine learning-trained models using the selected predictors have very good predictive skills, accounting for as much as 86% and 72% of the daily MSA and nss-SO4$^=$ variances ($R^2$), respectively.

4. Why do you only use the four machine-learning models to predict MSA and sulfate? Please explain the reason. To the best of my knowledge, decision tree model and deep learning might show the better performance compared with ANN and SVM.

We refer to Section 3.2 (in the revised manuscript) where we clarified that the most common types of ML algorithms have been trained under different advanced options and optimizations which can increase the performance and resilience of the algorithms. Following the reviewer's suggestion, we extended the ML models to include the Decision Tree (DT) type. Even, after considering this new ML model, the best performing model is still the GPR; indeed, the DT algorithm provided the lowest performance on our dataset. The panels below have been added to Figure 3 (for MSA) and Figure 4 (for nss-SO4$^=$), respectively.

[Figure]

[Figure]

The following paragraph has been inserted in the revised manuscript and the subheadings numbers and the text have been modified.

*"3.2.2 Decision Tree (DT)*

*The DT model is a non-parametric, non-linear model that generates a structure resembling a tree for classification and regression (Kotsiantis, 2013; Quinlan, 1986). It repeatedly divides the dataset into smaller subsets based on independent features from the input dataset. The split seeks to reduce variability within each group while increasing the variance between subsets. The final tree is made up of decision and leaf nodes. The decision node represents a condition on an attribute, and its branches indicate the conditions' outcomes. For additional information on DT, the reader is directed to [https://www.mathworks.com/help/stats/fitrtree.html](https://www.mathworks.com/help/stats/fitrtree.html). The critical parameter in this technique is determining when to terminate the dividing process. In this study, we set up three different minimum leaf sizes (minimum samples to split) to control the number of data that should be in the sub-branch to continue the splitting process, namely 4 (fine tree), 12 (medium tree), and 36 (coarse tree) as seen in Table 1."*

We inserted a new table (Table 1) to summarize the ML models used which could be immediate for the readers. Tables S1 and S2 have been merged as one table and present the evaluation measures, accordingly. Lastly, we refer to the use of neural networks in the manuscript which represents the deep learning models. Deep learning is a subtype of machine learning that resembles a neural network with three or more layers.

| Model Type | Preset | Hyperparameters if any |
|---|---|---|
| Support Vector Machines | Linear | |
| | Quadratic | |
| | Cubic | |
| | Fine Gaussian | Kernel scale = 0.61 |
| | Medium Gaussian | Kernel scale = 2.4 |
| | Coarse Gaussian | Kernel scale = 9.8 |
| Decision Tree | Fine | Minimum leaf size = 4 |
| | Medium | Minimum leaf size = 12 |
| | Coarse | Minimum leaf size = 36 |
| Regression Ensemble | Boosted | Minimum leaf size = 8 Number of learners = 30 |
| | Bagged | Minimum leaf size = 8 Number of learners = 30 |
| Gaussian Process Regression | Squared Exponential | |
| | Matern 5/2 | |
| | Exponential | |
| | Rational Quadratic | |
| Neural Networks | Narrow | Number of fully connected layers = 1 First layer size = 10 |
| | Medium | Number of fully connected layers = 1 First layer size = 25 |
| | Wide | Number of fully connected layers = 1 $1^{st}$ layer size = 100 |
| | Bi-layered | Number of fully connected layers = 2 $1^{st}$ layer size = 10 $2^{nd}$ layer size = 10 |
| | Tri-layered | Number of fully connected layers = 3 $1^{st}$ layer size = 10 $2^{nd}$ layer size = 10 $3^{rd}$ layer size = 10 |

Table 1: List of machine learning models used in the present study.

5. Section 4.6, I think the discussion about the spatiotemporal variations of MSA and sulfate seems to be very superficial and I suggest the authors should add more in-depth analysis.

We appreciate the reviewer's suggestion. In fact, we are focusing on many aspects of the proposed dataset that we believe will provide interesting scientific findings in future publications. Nevertheless, we necessarily need to limit the data exploitation in the present manuscript to avoid exceeding the scopes of ESSD, which is a data journal.

Anyway, in the revised version, we extended the analysis of this section to include more detailed information about the data distribution. The main modifications are:

- The Section title has been renamed to "Spatial distributions of MSA and nss-$SO_4^=$" instead of "Monthly MSA and $SO_4$ distributions"
- The annual spatial distributions (Figure 8 in the revised manuscript) have been added and the main spatial features has been explained.
- Figure 8 and Figure 9 (old version) have been merged as one figure (Figure 9 in the revised manuscript), to better and immediately compare the monthly variations.
- A new table (Table S2 in the revised supplementary) summarizing the statistics of the annual and monthly climatology (1998-2022) of MSA, nss-$SO_4^=$ and MSA:nss-$SO_4^=$ has been inserted.
- A new figure has been inserted (Figure S10 in the revised supplementary). It presents the spatial distribution of the monthly coefficient of variation (COV) calculated as the percentage of standard deviation divided by the mean, to evaluate the monthly stability of MSA and nss-$SO_4^=$. Higher COV indicates lower stability (many more variants).
- Accordingly, the main text has been modified to include the new analyses.

We refer to the tracked version of the revised manuscript where the modifications have been evidenced.

References

Kotsiantis SB. Decision trees: a recent overview. Artificial Intelligence Review 2013; 39: 261-283.

Mansour K, Decesari S, Ceburnis D, Ovadnevaite J, Rinaldi M. Machine learning for prediction of daily sea surface dimethylsulfide concentration and emission flux over the North Atlantic Ocean (1998–2021) Science of The Total Environment 2023; 871.

Quinlan JR. Induction of decision trees. Machine Learning 1986; 1: 81-106.

Zhou SQ, Chen Y, Wang FH, Bao Y, Ding XP, Xu ZJ. Assessing the Intensity of Marine Biogenic Influence on the Lower Atmosphere: An Insight into the Distribution of Marine Biogenic Aerosols over the Eastern China Seas. Environmental Science & Technology 2023.

---

## Author Comment (AC2)

**Answers to Referee 2**

We would like to express our gratitude to the reviewers for their insightful comments on the manuscript. According to the comments, the manuscript has been revised. Please find our replies in the pdf file. In the responses, the reviewer's comments are in black text, and our responses are in blue and *the main text modifications to the revised manuscript are in italics*.

The study of Mansour and colleagues represents a step forward towards the prediction of biogenic sulfur in aerosols, which has climatic and geochemical importance. The authors used several machine learning approaches, each with alternative configurations, to estimate the concentration of the two major atmospheric oxidation products of plankton-made dimethyl sulfide: non-sea-salt sulfate and methanesulfonate. Finally, the best performing model was used to produce daily gridded datasets for these compounds over the North Atlantic Ocean. I found the study methodologically robust and well written, but some issues should be addressed before publication.

General comments

I suggest using nss-SO4, not just SO4, throughout. Abbreviating nss-SO4 may confuse readers because, unlike MSA, SO4 has large anthropogenic and volcanic sources. The same applies to MSA:nss-SO4 ratios.

We agree with the reviewer. The abbreviation has been changed throughout the manuscript (nss-SO$_4^=$ instead of SO$_4$), including the figure axes labels/ captions and table headings/ captions. In the revised manuscript, the following clause has been eliminated:

*"Throughout the present study, we abbreviate the nss-SO$_4^{2-}$ concentration as SO$_4$ and MSA concentration as MSA, for simplicity."*

L141: Please provide a quantitative comparison between nss-SO4 and the non-refractory SO4 pool measured with the HR-ToF-AMS, e.g. an indication of the mean absolute and/or relative difference between the two estimates. Just stating they are "approximately equivalent" is not very reassuring. Can the authors exclude the possibility that, in some instances, significant proportions of nss-SO4 are in aerosol fractions not captured by the HR-ToF-AMS?

Our statement is based on the general understanding of the AMS measuring principle. Indeed, the AMS is very sensitive to sulfate in the form of ammonium sulfate, ammonium bisulfate and sulfuric acid (Chen et al., 2019; DeCarlo et al., 2006), which are the main forms under which nss-$SO_4^=$ is present in marine aerosol (Ovadnevaite et al., 2014). Therefore, the possibility that the HR-ToF-AMS may underestimate the nss-$SO_4^=$ concentration by missing some fraction of it, is highly unlikely. Conversely, sea-salt is considered a refractory component for the AMS, which means that sea-salt components tend to evaporate inefficiently within the AMS oven. This, together with the small contribution of sulfate in sea-salt (only 7.7% in mass) and considering the size distribution of sea-salt, that mostly falls outside the operative range of the AMS, makes the contribution of sea-salt-sulfate in AMS measurements usually negligible. In any case, in principle it may be possible to have an overestimation of the nss-$SO_4^=$ in case of high sea-salt-$SO_4^=$ contribution. Ovadnevaite et al. (2014) quantified these cases, concluding that a non-negligible contribution of sea-salt-$SO_4^=$ in the MHD database can be observed only for cases of low sulfate and high sea-salt concentrations associated to high wind speed events during winter months, when the contribution of sulfate is in any case close to the detection limit and negligible with respect to the high biological activity period. Similarly, Saliba et al. (2020), presenting the NAAMES database, states that non-refractory-$SO_4^=$ (measured by AMS) excludes refractory particles that likely contain the majority of sea-salt sulfate and that it is therefore approximately equivalent to nss-sulfate. To support this statement in a more quantitative way, we compared the concentrations of sea-salt and sulfate reported by Saliba et al. (2020), assuming a 7.7% $SO_4^=$ contribution in sea-salt: only during the winter cruise the contribution of sea-salt-$SO_4^=$ to the total AMS-$SO_4^=$ signal is non-negligible (54%), while in the other cruises it is around 10%. Anyway, this estimate is biased by the different cut-off of the samples used for sea-salt analysis (1.1 µm) and the AMS (~0.8 µm), which makes these numbers very likely overestimated.

Finally, if the reviewer is instead worried about the potential presence of nss-$SO_4^=$ in particles larger than the AMS upper cut-off, this may be true [and maybe even more for MSA (Rinaldi et al., 2011)] but we clearly state in the manuscript that our dataset refers to submicrometer particles, which falls in the size range of AMS and which are the more relevant climatically.

The authors use HYSPLIT driven by the Global Data Assimilation System (GDAS1) (1° × 1°) of the National Centers for Environmental Prediction (NCEP) to calculate back-trajectories (section 2.3). A different reanalysis, ERA5, is used to obtain meteorological predictor variables for machine learning methods (section 2.5), as well as the BLH used to analyze HYSPLIT-derived back trajectories (section 3.1.1). Can the use of different reanalyses in different parts of the study introduce inconsistencies?

In this study, the GDAS1 data set is only utilized to generate the Back-trajectories data, as one of the archived datasets in the HYSPLIT system. To achieve high spatial resolution (0.25° × 0.25°) of biogenic sulfur aerosol concentrations, we use the ERA5 dataset as predictors in machine learning models. The BLH was extracted along BTs in the same way as other atmospheric predictors because it also serves as a predictor. Indeed, all predictors included in model training were retrieved in the same way, therefore we do not believe such a strategy will create errors or uncertainties in the current study.

Section 3.3: please consider reporting other metrics, like the Prediction-Observation linear slope (which would be 1 for perfect model predictions) -- OK, this is shown in Fig. 6 and 7. Just consider introducing this metric in section 3.3.

We added the slope value as a metric to evaluate ML models in Figures 3 and 4 too. The following clause complements Section 3.3 in the revised manuscript.

*"The predicted-observed linear slope is the last metric used to evaluate the performance of ML models. It determines the rate of change of the predicted variable concerning the observed variable and should be close to unity for skilled model predictions."*

We modified panels (a-b) of Figure 6 and panels (a-d) of Figure 7 to make observations on the x-axis that are consistent with the explanation given above.

L272: How can this procedure prove causal relationships?

We agree with the reviewer that the sentence is misleading. We rephrased the sentence in the revised manuscript by eliminating the first part, now it reads:

*"We used multilinear regression to assess the contribution of each predictor to MSA and nss-SO$_4^=$ variations."*

Specific

L25, L85…: "constructed" >> "reconstructed"

Corrected.

L27: what is the "ensemble" ML method? OK, later defined as "regression ensemble"

We also added the word "regression ensemble" instead of "ensemble" in the specified line.

L42: marine phytoplankton >> marine microbes (phytoplankton are not the only DMS producers)

marine microbes replaced marine phytoplankton.

L49: elevated temperature and solar radiation >> elevated temperature OR solar radiation

Corrected.

L101 and paragraph: please revise whether the AMOC is the phenomenon you actually want to highlight here. Perhaps a mention to the Gulf Stream and the North Atlantic Current is enough (which indeed are components of the much wider phenomenon termed AMOC).

We agree to the reviewer's suggestion and highlighted the Gulf Stream instead of AMOC. The paragraph has been updated as follows:

*"The key climate-relevant features in the study domain are the Gulf Strem, its northern extension towards Europe known as the North Atlantic Current (NAC), and the cyclonic subpolar gyre (SPG) (Rhein et al., 2011). The Gulf Stream is a warm Atlantic Ocean flow that begins in the Gulf of Mexico and moves through the Straits of Florida before continuing up the eastern coast of the United States (Buckley and Marshall, 2016). These warm northward-flowing waters meet the cold southward-flowing waters of the Labrador Current and the western boundary current of the cyclonic subpolar gyre, ultimately turning east and heading toward Northwest Europe as the NAC. The NAC then splits into multiple branches that enter the subpolar gyre, one of which passes via the Iceland Basin and the other through the Rockall trough (Fratantoni, 2001). The NA SPG extends from 45° N to around 65° N and comprises the sills between Greenland, Iceland, the Faroe Islands, and Scotland. Such circulation phenomena are crucial for the modulation of the temperate climate of north-western Europe (Marzocchi et al., 2015), and the dynamics of SPG determine the rate of deep and intermediate water formation (sinking dense and cold surface waters through air-sea heat exchanges in the wintertime) particularly in the Labrador Sea (Katsman et al., 2004).*

*Accordingly, they contribute to the regional changes of primary production and the subsequent biogenic emissions in the study domain."*

L238: were predictors averaged with or without the weighting factor e^(-t/72) used to compute R_0 and R_B? it would make sense to apply this weighting when using the meteorology along the BTs as predictor.

Thank you for pointing this out. We compared the weighted average $F_{DMS}$ and SRF along BTs (as used in the present manuscript) to the same values when the weighting factor e^(-t/72) is included. The results reveal a strong connection between them (*r = 0.99* for $F_{DMS}$ and *r = 0.98* for SRF), as shown in the following scatter plots.

[Figure]

(Left) Comparison of $F_{DMS}$ values with and without incorporating the weighting factor. (Right) Comparison of SRF values with and without incorporating the weighting factor.

In addition, we compared the $F_{DMS}$/ SRF (with and without the weighting factor) and MSA measurements at Mace Head. The results (figures below) show that incorporating the weighting factor does not change the relationship between predictors (*e.g.,* $F_{DMS}$ & SRF) and response (*e.g.,* MSA). This may be due to the fact that the removal of submicron aerosol particles is negligible over a 1–3 days transport time. As a consequence, for this study, whether or not incorporating this weighting factor does not have a significant impact on the analysis results, we retained using the weighted mean along BTs without including the weighting factor.

[Figure]

Scatter plots between $F_{DMS}$ at the selected marine air masses and the in-situ observed MSA concentrations at Mace Head.

[Figure]

Scatter plots between SRF at the selected marine air masses and the in-situ observed MSA concentrations at Mace Head.

L272: was MLR applied to untransformed or log-transformed data (as done for the correlation analysis)?

Yes, it was. We clarified this point in the caption of the multilinear regression table (Table 4; in the revised manuscript).

Typos

L232: "NAAMEAS" cruises

Corrected.

L477: "Quantitively"

Corrected.

L529: southern >> southward

Corrected.

References

Buckley MW, Marshall J. Observations, inferences, and mechanisms of the Atlantic Meridional Overturning Circulation: A review. Reviews of Geophysics 2016; 54: 5-63.

Chen YL, Xu L, Humphry T, Hettiyadura APS, Ovadnevaite J, Huang S, et al. Response of the Aerodyne Aerosol Mass Spectrometer to Inorganic Sulfates and Organosulfur Compounds: Applications in Field and Laboratory Measurements. Environmental Science & Technology 2019; 53: 5176-5186.

DeCarlo PF, Kimmel JR, Trimborn A, Northway MJ, Jayne JT, Aiken AC, et al. Field-deployable, high-resolution, time-of-flight aerosol mass spectrometer. Analytical Chemistry 2006; 78: 8281-8289.

Fratantoni DM. North Atlantic surface circulation during the 1990's observed with satellite-tracked drifters. Journal of Geophysical Research-Oceans 2001; 106: 22067-22093.

Katsman CA, Spall MA, Pickart RS. Boundary current eddies and their role in the restratification of the Labrador Sea. Journal of Physical Oceanography 2004; 34: 1967-1983.

Marzocchi A, Hirschi JJM, Holliday NP, Cunningham SA, Blaker AT, Coward AC. The North Atlantic subpolar circulation in an eddy-resolving global ocean model. Journal of Marine Systems 2015; 142: 126-143.

Ovadnevaite J, Ceburnis D, Leinert S, Dall'Osto M, Canagaratna M, O'Doherty S, et al. Submicron NE Atlantic marine aerosol chemical composition and abundance: Seasonal trends and air mass categorization. Journal of Geophysical Research-Atmospheres 2014; 119: 11850-11863.

Rhein M, Kieke D, Huttl-Kabus S, Roessler A, Mertens C, Meissner R, et al. Deep water formation, the subpolar gyre, and the meridional overturning circulation in the subpolar North Atlantic. Deep-Sea Research Part Ii-Topical Studies in Oceanography 2011; 58: 1819-1832.

Rinaldi M, Decesari S, Carbone C, Finessi E, Fuzzi S, Ceburnis D, et al. Evidence of a natural marine source of oxalic acid and a possible link to glyoxal. Journal of Geophysical Research-Atmospheres 2011; 116.

Saliba G, Chen CL, Lewis S, Russell LM, Quinn PK, Bates TS, et al. Seasonal Differences and Variability of Concentrations, Chemical Composition, and Cloud Condensation Nuclei of Marine Aerosol Over the North Atlantic. Journal of Geophysical Research-Atmospheres 2020; 125.

---

## Author Response (AR2)

**Answers to Reviewers**

We would like to express our gratitude to the editor and the reviewers for their positive feedback on the manuscript. We revised the manuscript after considering the minor comments raised. In the responses below, the reviewer comments are in black text, and our responses are in blue and *the main text modifications to the revised manuscript are in italics*.

**Reviewer 2**

The paper has been significantly improved after incorporation of the reviewer's suggestions. My main concerns have been satisfactorily addressed. Therefore, I support publication after minor revisions. Below I list additional minor comments. I prompt the authors to incorporate them to further improve the manuscript:

1) The authors could mention the importance of reconstructing atmospheric MSA fields and its sources. MSA is widely used tracer of marine biological productivity. In the North Atlantic subpolar gyre, MSA from ice cores (deposited from the atmosphere) has been used to reconstruct decadal to centennial scale variability of marine productivity. An appropriate citation is Osman et al. 2019 Nature (https://www.nature.com/articles/s41586-019-1181-8).

We thank the reviewer for pointing out this valuable study. We modified the 1st paragraph in the introduction Section to read as:

*"As a result, biogenic sulfur aerosols play a central role in ocean-atmosphere interactions and regional climate change, and it is critical to parameterize and characterize biogenic MSA and nss-$SO_4^=$ across different sea areas and identify their sources to constrain the past, current and future climate impacts of both species (Hodshire et al., 2019; Gondwe et al., 2003). For instance, MSA observations from Greenlandic ice cores have been used to study the variability of subarctic Atlantic Ocean productivity from decadal to centennial time scales (Osman et al., 2019)."*

2) The inclusion of the model:obs slope metric is very welcome. This metric is typically around 0.80 on a log-log scale. Thus, even though the ML approaches used in this study significantly improve the prediction of nssSO4 and MSA in comparison to previous studies, they still tend to overestimate low

extremes and underestimate high extremes, as clearly seen in several figures (3, 4, 6 and 7). This is a common issue in statistical models. I prompt the authors to add a cautionary note on this source of uncertainty, which limits our quantitative understanding of extreme emission events.

We rephrased the 3rd paragraph in Section 4.1 to read as:

*"Importantly, the implemented ML models can reconstruct MSA and nss-$SO_4^=$ daily time series characteristics with remarkable consistency between observed and predicted data, except for extremely high and low concentrations. This is mostly due to the low probability of such concentrations in the observed dataset, which inhibits ML models from reconstructing them. The quantitative comprehension of exceptional emission extremes is not addressed in this study; nonetheless, their occurrence and possible implications deserve to be investigated in future studies."*

3) It appears that the term "error" should be replaced by "deviation", i.e., replace RMSE and MAE by RMSD and MAD, because observations have their own uncertainties. Strictly speaking, error should be used only when truth is known (.e.g in theoretical studies).

We agree with the reviewer on the "philosophical" point of view that observations have their uncertainties and do not necessarily represent the truth. On the other hand, in the manuscript we made use of the standard parameters and terminology used in a number of papers where model outputs are validated against observations. We prefer to keep these terms as they are (MAE and RMSE) to be consistent with the vast majority of the literature and to avoid ingenerating confusion in the readers about the main statistical terms. Indeed, the mean absolute deviation (MAD) and the root mean square deviation (RMSD) are used to measure the deviation or variability in a dataset from its mean value. For example, MAD is calculated as the absolute difference between each data point and the population mean, then, the average of these absolute differences is obtained. Indeed, these measures are not possible in our case.

4) Please, review thoroughly the references to pick potential typos or minor spelling errors.

Reviewed and checked.